# The Legacy of Einstein's Eclipse, Gravitational Lensing

**Jorge L. Cervantes-Cota** [1] , **Salvador Galindo-Uribarri** [1] **and George F. Smoot** [2,3,4,*]

[1] Department of Physics, National Institute for Nuclear Research, Km 36.5 Carretera Mexico-Toluca, Ocoyoacac, C.P.52750 Mexico State, Mexico; jorge.cervantes@inin.gob.mx (J.L.C.-C.); salvador.galindo@inin.gob.mx (S.G.-U.)

[2] IAS TT & WF Chao Foundation Professor, Institute for Advanced Study, Hong Kong University of Science and Technology, Clear Water Bay, Kowloon 999077, Hong Kong

[3] Université Sorbonne Paris Cité, Laboratoire APC-PCCP, Université Paris Diderot, 10 rue Alice Domon et Leonie Duquet, CEDEX 13, 75205 Paris, France

[4] Department of Physics and LBNL, University of California, MS Bldg 50-5505 LBNL, 1 Cyclotron Road, Berkeley, CA 94720, USA

\* Correspondence: gfsmoot@lbl.gov; Tel.: +1-510-486-5505

**Abstract:** A hundred years ago, two British expeditions measured the deflection of starlight by the Sun's gravitational field, confirming the prediction made by Einstein's General Theory of Relativity. One hundred years later many physicists around the world are involved in studying the consequences and use as a research tool, of the deflection of light by gravitational fields, a discipline that today receives the generic name of Gravitational Lensing. The present review aims to commemorate the centenary of Einstein's Eclipse expeditions by presenting a historical perspective of the development and milestones on gravitational light bending, covering from early XIX century speculations, to its current use as an important research tool in astronomy and cosmology.

**Keywords:** gravitational lensing; history of science

**PACS:** 01.65.+g; 98.62.Sb

## 1. Introduction

On the 6th of November of 1919 the results of two British expeditions to observe a solar eclipse that took place on May 29th of that same year, were released at a joint meeting of the Royal Society and the Royal Astronomical Society. The purpose of the expeditions was "to determine what effect, if any, is produced by a gravitational field on the path of a ray of light traversing it" [1].

The conclusion given at that meeting was: "the results of the expeditions to Sobral and Principe can leave little doubt that a deflection of light takes place in the neighborhood of the Sun and that it is of the amount demanded by Einstein's generalized theory of relativity, as attributable to the Sun's gravitational field" [1].

Exactly one hundred years have passed since Sir Arthur Stanley Eddington, Sir Frank W. Dyson and Charles Davidson measured the apparent shifting of stars locations near the limb of the Sun during a solar eclipse. To commemorate the centennial of Eddington's and colleagues' key observation that tested Einstein's theory, shaking our conceptions of space and time, we present a historical review of several phenomena collectively known as gravitational lensing effects, that have their roots in those observations made in 1919.

Mass bends light, the gravitational field of a massive stellar object causes light rays passing close to the object to bend and focus somewhere else. If the light source is at a right distance and bright enough, and if the interposed object (or objects) is sufficiently massive and close to the line of sight,

the gravitational field of the interposed object acts like a lens, focusing the light and producing effects such as: magnification of sources, image distortions, replicating images of a single source or shifting the apparent location of the source. These four consequences of gravitational light deflection are collectively known as gravitational lensing effects or lensing for short.

In recent decades observers have found that lensing effects offer the opportunity to study background sources, some of them intrinsically weak and distant that would not be observable without lensing. Awareness of lensing intervenes in estimation counts of some of these background sources such as quasi-stellar objects (quasars) as well as on estimates of flux variability of extragalactic compact objects. In addition, lensing effects are a cosmological and astrophysical tool providing unique insights into the nature of the foreground lens, such as its mass distribution or its nature (dark matter candidates) or the determination of the Hubble parameter. Lensing applications cover a wide variety of topics, among them it has been used in the search of exoplanets or to resolve structures of active galaxies. Gravitational lensing has been applied to the entire electromagnetic spectrum from radio to gamma rays, and recently using gravitational waves, allowing us to study structures as small as black holes and as large as galaxy clusters.

For years little attention was paid to lensing, but presently many are flocking to the field. Today lensing has become, in its own right, a very dynamic research tool. Even though modern-day lensing is still a young field, it has a historical background dating back to a little more than two centuries as it is naturally associated to the development of ideas on gravitational light bending. At the same time, the interest in gravitational bending of light originates in early speculations on the existence of massive stellar objects ("dark stars") capable of modifying trajectories of light rays. This led, at that time, the need to value the mass of stars and the development of means to do so, such as the torsion balance. This eventually brought Henry Cavendish, through a series of episodes to try to estimate the degree of deflection that the Sun exerts on a ray of light that passes in its vicinity.

In this review, we shall start by giving an account of little-known episodes on these matters occurring at the end of the XVIII and early XIX century. There followed a period of one hundred years where no evidence appears that someone has tackled the idea again until Einstein recognized that the gravitational field of our Sun would be strong enough to produce a measurable effect on the bending of light by its own field. Later we will talk about the attempts to prove the correct value of this deflection angle made by Einstein, before he developed General Relativity (GR)[1] and after. Next, we will speak about the prediction of lensing effects made by some visionaries before the first observation that would effectively confirm the existence of lensing in 1979, barely 40 years ago.

The discovery of the first lensed quasar in 1979 raised some interest in using gravitational lensing for the investigation of both the nature of the lenses as well as the sources. But then, a number of lensed systems detections within few years after the first sighting, triggered a prompt expansion in the amount of publications in this field at very high rates. Therefore, at this point we will abandon our chronological storyline to follow the narrative into several topics in separate sections, covering the relevance of lensing in astronomical observations and its cosmological implications.

## 2. Early Thoughts on the Influence of Gravity on Light

One of the first mentions of a massive object capable of bending the trajectory of a light ray due to its strong gravitational field was made by Erasmus Darwin (1731–1802), grandfather of Charles. Erasmus was a famous physician in his time, a great naturalist and a polymathic genius. He was also a poet and a scientific popularizer. His poems evoked man's scientific endeavors, as he believed that the reading of his poetry by the general public would awaken their curiosity on nature. In 1791 he published a book of verses entitled "*The Botanic Garden*", whose aim was mainly to publicize his startling scientific beliefs. His book explored everything: from meteors, clouds and coal, to shell-fish

---

[1]   For recent overviews of GR and recent developments, see, e.g., [2,3] and references therein.

and steam-engines. Erasmus' book of poems is divided into two parts. In the first part, entitled "Economy of Vegetation" we can read [4]:

> *Star after star from heaven's high arch shall rush,*
> *Suns sink on suns, and systems systems crush,*
> *Headlong, extinct, to one dark centre fall,*
> *And death and night and chaos mingle all!*
> (*Economy of Vegetation*, 1791, Canto IV)

In the first couple of lines of the excerpt, Darwin exposes the conjecture made by the astronomer William Herschel (1738–1822), that the stars grouped in a globular cluster will approach each other until they finally merge in a huge mass. This conjecture was published by Hershel in 1785 in the Philosophical Transactions of the Royal Society and is cited as a footnote in the same poem by Darwin [5]. The last pair of lines in the excerpt refers to one sort of "dark centre". Here Darwin exposes the proposal of another of his well-known friends, the Reverend John Michell (1724–1793) whose reasoning on the existence of such "dark" objects is more elaborate and implies further explanation.

During the 1770's decade Michell was trying to investigate how much of its mass, our Sun was releasing by its own emission of light. For Michell and some of his contemporaries, a disturbing problem at that time was the question, arising from their belief on the corpuscular theory of light, of why the Sun did not deplete itself by its supposed vast outpouring of mass. In order to measure that supposed loss of mass, Michell invented a torsion balance. From one of the balance arms, Michell placed a very thin copper plate hanging from a slender harpsicord wire and from the second arm a counterweight. The apparatus was encased inside a box to prevent being disturbed by any motion of the air. By means of a two feet diameter concave mirror, sunlight was focused on the copper plate. By observing the angular displacement of the balance by the impulse of the sunrays impinging on the plate, Michell proposed to measure the sunlight impulse and thus "the quantity of matter contained in the rays of the Sun" [6].

When performing the experiment, Michell focused sunlight on the plate and the balance arms apparently turned a certain angle that was written down. The same procedure was to be repeated three to four times and the experiment was to be done again, but this time focusing sunlight on the back side of the thin copper plate. Regrettably when the experiment was again repeated but this time on the back of the copper plate, the plate began to change its shape under the extreme heat provoked by exposure to sunlight and finally it melted down. Michell's efforts are described in Joseph Priestley book "*The History and Present State of Discoveries Relating to Vision, Light and Colours*" (1772). Nevertheless Joseph Priestley (1732–1804) used Michell's "measurements" to estimate that the Sun was losing, to everybody's great relief, just less than grain of weight (64.79891 milligrams) per day [6].

Reverend Michell's activity on scientific research did not end with the melting of his torsion balance. By 1783 a thought came to him while considering a hypothetical method to determine distances and weights of "fixed stars". Michell's method was based on a corollary of Newton's Principia, which indicates how to discern the mass of a planet in terms of the mass of the Earth by knowing their orbits radii and their periods of revolution [7]. Michell's idea was to extrapolate this well-known corollary to binary star systems. But here's the rub, the problem with the method was that binary star systems had yet to be discovered, as their existence had only been conceived in Michell's mind. Nevertheless, Michell argued that the possibility of existence of these binary stars systems was very high because Herschel had already discovered large amounts of double stars and he speculated that maybe some of them would be sufficiently close so that their mutual attraction would bind them together forming the binary system. In particular, the hypothetical binary star system that Michell had in mind should consist of a central massive star with a lighter orbiting star, so if astronomers could determine the orbital radius of the satellite star and its period of revolution, the mass of the central star could be estimated in terms of solar masses.

On the other hand, Michell reasoned that light corpuscles, emerging from the surface of the central star, would have their speed reduced by the star's gravitational attraction according to the star's mass (this idea is in principle similar to what is now known as gravitational redshift, which refers to the loss of energy). So by measuring the speed of light reduction from the central star the distance to that luminary could be determined.

Consequently, a fundamental measurement of Michell's method was the evaluation of the speed decrement of the light emitted by the star. Based upon Newton's hypothesis that the faster light travels, the less it is refracted by a medium, Michell believed that by using an achromatic prism, an observed increment in the angle of diffraction for a slower light ray would provide the precise measurement of the light's velocity reduction. The full-length method was proposed by Michell in a letter dated 26 May 1783, send to Henry Cavendish (1731–1810) to be communicated to The Royal Society (read 27 November 1783) [8].

In the course of his reasoning, Michell speculated in the same letter, that a stellar mass might be so large that the speed magnitude of an emitted light corpuscle could well be lesser than the corresponding escape velocity for the star mass and therefore corpuscles could not escape from that "dark star". In his 1783 paper it can be read [8]:

> "...*If the semi-diameter of a sphere of the same density with* [as] *the Sun were to exceed that of the sun in the proportion of 500 to 1,... and consequently, supposing light to be attracted by the same force in proportion to its vis inertiae* [mass], *with other bodies, all light emitted from such a body would be made to return towards it, by its own proper gravity*".

In other words, John Michell, in the eighteenth century conceived the possibility of the existence of a "Newtonian" black hole, and what is more, in the same line of thought as the significance of the Schwarzschild radius, he estimated the minimum radius (semi-diameter) that a dark star should have so its light does not escape (500 times the Sun radius). Michell went further to suggest one way of detecting Newtonian black holes:

> "...*yet, if any other luminous bodies should happen to revolve about them* [dark stars] *we might still perhaps from motions of these revolving bodies infer the existence of the central ones...*"

Here it is fair to mention that independently in 1796, almost thirteen years later, the illustrious French mathematician, astronomer and physicist Pierre-Simon Laplace (1749–1827), in his famous work "*Exposition du Système du Monde*", considered the existence of "*corps obscures*", a similar concept to that of Michell [9]. However, Laplace deleted the concept of "*corps obscures*" in later editions of his book from the third edition (1808) onwards, in view perhaps of judging the idea highly speculative [10].

Returning to the aforementioned letter (26 May 1783) of Michell to Cavendish, the former conceded that his proposed method required the "concurrence of so many circumstances" (for a start, discovering binary star systems). But as it happens in "good stories", just at that time (May 1783) John Goodricke (1764–1786) a profoundly deaf amateur astronomer submitted a paper to Philosophical Transactions on discovered variations in the luminosity of the star Algol (Beta Persei) [11]. Goodricke found that the dimming of Algol's light occurred exactly every 2.767 days and speculated that a possible explanation could be an "interposition of a large body revolving around Algol" (what today is called an eclipsing binary). This might have prompted Michell to think that such "large body" could well be a dimmer star eclipsing a brighter central star [12]. This finding drove Michell to write a letter to Cavendish (2 July 1783) expressing that, the efficiency of his method "almost with certainty", would be soon confirmed [13].

Cavendish in turn encouraged, William Herschel and the Royal Astronomer Nevil Maskelyne (1732–1811), the two most prestigious astronomers of England at that time, to perform astronomical observations to try to prove Michell's method. Both astronomers made several observations without being able to find any evidence that could confirm the feasibility of Michell's method. After several unsuccessful observations, Cavendish concluded that "there is not much likelihood of finding any

stars whose light is sensibly diminished" [14]. In the end Michell went on to acknowledge Cavendish verdict that there might be no stars out there "big enough" [15].

## 3. First Estimates of the Effect

When John Michell died ten years later in 1793, his instruments, including his torsion balance, were given to Queens' College, Cambridge. Eventually his balance passed into the hands of his lifelong friend Henry Cavendish, who perfected the apparatus and first performed in 1798 the experiment now known as the Cavendish Experiment.

But perhaps what Michell also bequeathed Cavendish was the disposition to find some way to test the influence of gravity on light. The significant exchange of letters on the subject between Michel and Cavendish could very well predispose the latter to continue with the attempt. At some point in his life, Cavendish must have focused his attention on the other effect that gravity produces on corpuscles of light, that is to say, bending their trajectories rather than (the supposed) slowing their speeds. Cavendish interest in the bending of light probably surged during a late period of his life when he devoted himself to the study of trajectories of comets by action of gravity.

Unfortunately, Cavendish often passed up publishing his work and many of his findings were not even disclosed to his colleagues. Cavendish did not write a single book and published in his life less than 20 papers. He lived in an era when the "publish or perish" paradigm was unheard. Nevertheless, after perishing he left abundant unpublished notes.

In 1922 some of Cavendish unpublished astronomical papers where examined by the Astronomer Royal, Sir Frank W. Dyson (1868–1939). Among the numerous papers Sir Frank found an isolated scrap of paper. Dyson transcription is shown below [16],

> *To find the bending of a ray of light which passes near the surface of any body by the attraction of that body: Let s be the centre of body and "a" a point of surface. Let the velocity of body revolving in a circle at a distance as from the body be to the velocity of light as 1:u, then will the sine of half bending of the ray be equal to* $1/(1 + u^2)$.

Immediately below Dyson added a comment: "[*This deflection is half the amount given by Einstein's law of gravitation*]". In effect, in 1987 Clifford Will made a detailed analysis on the deflection of light by a body based in Newtonian Gravitation and the corpuscular model of light [17]. In his analysis he found that the sine of half bending of the ray mentioned by Cavendish is $\sin \delta = 1/e$, where $e$ is the eccentricity of the hyperbolic orbital path of the corpuscular light ray which turns out to be, $= 1 + \frac{R\,c^2}{G\,m}$, where $G$ is the gravitational constant, $m$ is the mass of the attracting body, $c$ the speed of light and $R$ is the distance of the light ray closest approach to $m$. Consequently, the symbol $u$ employed by Cavendish is given by $u = \left(\frac{R\,c^2}{G\,m}\right)^{\frac{1}{2}}$. Therefore the Newtonian result for the total deflection is, $\theta_{Newton} = 2\frac{G\,m}{R\,c^2}$, which is, as correctly pointed by Dyson in his footnote, half of the amount given by Einstein's GR.

It is unknown when Cavendish wrote this note as there is no date written on it, but according to Jungnickel, and McCormmach, the watermark on the sheet where the note was written reads "1802" [18].

But it was not until 1804 when the first published article that deal with the deflection of light by a body, appeared in a major astronomical journal. The paper, "On The Deflection Of Light Ray From Its Straight Motion Due To The Attraction Of A World Body Which It Passes Closely," was authored by a Bavarian astronomer working at the Berlin Observatory named Johann George von Soldner (1776–1833) [18]. Soldner's procedure was similar to that of Cavendish treating light as ordinary particles that move in Newton's gravitational field. Soldner's result for the deflection angle agrees with that of Cavendish to a first order approximation [19]. Soldner inferred that a light ray close to the solar limb would be deflected by an angle of approximately 0.85 arc sec which corresponds to half the value of the actual deflection. Soldner's work fall into oblivion for more than 100 years until anti-Semitic fanatics brought it to light in 1921 to falsely claim that Albert Einstein had plagiarized it.

After the pair of isolated investigations by Cavendish and Soldner on light bending, no further studies on the topic were undertaken until the beginning of the 20th century. Possibly this came about as the wave conception of light was to supersede the corpuscular model due to the weight of experimental evidence built up during the earlier part of the 19th century.

## 4. Einstein Bends the Corner

It was not until 1907, when Albert Einstein had already completed the theory of Special Relativity, that Einstein considered again the problem on the influence of gravitation on light. That year he published a review paper on Special Relativity entitled "*On the relativity principle and the conclusions drawn from it*" [20]. It was in this work, in which he first introduced the equivalence principle, that he wrote: "*It follows from this that the light rays . . . are bent by the gravitational field*". Then he went on remarking: "*Unfortunately the influence of the Earth's gravitational field is according to our theory is so slight [. . . ] that there exists no prospect for a comparison of the results of the theory with experiment*" [20].

Einstein changed his mind in 1911 when he contemplated again the question of the influence of gravitation on light. That year he published an article entitled "On the influence of gravity on the propagation of light" [21]. This time he recognized that the gravitational field of our Sun would be strong enough to produce a measurable effect on the bending of light by its own field. In this paper Einstein obtained, by merely exercising the equivalence principle, the same result for the deflection angle that Cavendish and Soldner obtained a century before (i.e., 0.85″, see $\theta_{Newton}$). However, given that the theory of General Relativity (GR) had not been fully developed in its entirety, Einstein's result did not take into account the space-time curvature around the Sun. Therefore, his calculation using the equivalence principle alone was an approximation and a factor of two off the GR value.

## 5. Early Eclipse Observations

In August 1911, Leo Wentzel Pollack (1888–1964), an astronomer at the German University of Prague paid a visit to Royal Observatory (Königliche Sternwarte) in Berlin. Pollack had had several previous conversations with Einstein on the deflection of light by the Sun since at that time Einstein was his coworker as professor of theoretical physics at the same University.

On Pollack arrival at the Königliche Sternwarte in Berlin, a junior astronomer by the name of Erwin Finlay Freundlich (1885–1964) was commissioned to show Pollack the premises. During the course of this tour, Pollack mentioned Einstein's recent paper "On the influence of gravity on the propagation of light" and his petition to astronomers to test his predictions. Freundlich was mesmerized to hear what Pollack said on Einstein's prediction, because his job at the observatory was routine work and not intellectually challenging. Readily Freundlich wrote Einstein that same night offering his help on the possibility of observing the light deflection effect in the vicinity of the solar rim [22]. By the end of 1911 an epistolary correspondence interchange between Einstein and Freundlich was established [23].

Freundlich proposed Einstein the option of examining photographic plates of past solar eclipses to find out shifts on fixed stars positions of those located nearby the solar rim (Figure 1). Einstein got along with the idea. Freundlich commissioned himself to look for old photographic plates in his institution as well as to write to several observatories around the World requesting them to provide plates of past eclipses. His request did not have the expected response, and those plates that came available to him produced no results. However, the American astronomer Charles Dillon Perrine (1867–1951) director of the Argentine National Observatory answered positively and agreed to directly meet Freundlich to discuss the problem in person as he was going to assist to the "Carte du Ciel" meeting in Paris, to be held in the coming October 1911 [24]. After the Paris meeting Perrine would then visit the Pulkovo Imperial Observatory, located a few kilometers south of St. Petersburg. The train that would take Perrine from Paris to St. Petersburg was to stop for a few hours at Berlin, fact that both Perrine and Freundlich saw as the opportunity to meet there. During the meeting they discussed the intention to verify Einstein's calculation on light's deflection during a solar eclipse. Charles Perrine expertise helped in the discussion as he before had accompanied four eclipse expeditions and was

in charge of the one sent from Lick Observatory to Sumatra in 1901 before he was appointed in 1909 head of the Argentinian observatory. During the course of the conversation at Berlin, Freundlich asked Perrine his opinion on the possibility of analyzing older photographic plates previously obtained by Perrine in the occasion of his multiple past expeditions while working at Lick. In turn, Perrine expressed his doubts on the usefulness of past plates as the field of view was reduced and exposures were short [24]. From that meeting on, both astronomers started an open collaboration.

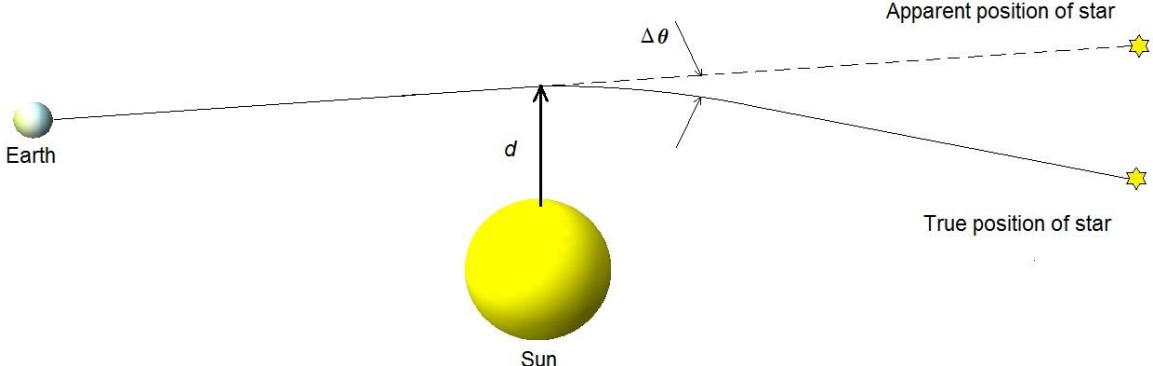

**Figure 1.** Light ray deflection of an angle $\Delta\theta$ by the Sun. The distance of the closest approach to the Sun is d.

In the eve of 1912, Freundlich sent a letter to Perrine, then back in the Argentine Observatory, requesting his observatory to organize a team to make the appropriate observations in the eclipse that would occur in Brazil on October 10 of that year, a proposal that Perrine willingly accepted [24]. On Thursday October 10 the Argentine team was prepared in Brazil to observe the eclipse, but "unfortunately" for the team of astronomers (and "fortunately" for Einstein since the numerical value of his prediction was wrong), that day a storm kept the sky covered preventing observations. It is interesting to remark that this was the first (unsuccessful) attempt to measure light bending during a solar eclipse.

Four days after the unsuccessful attempt in Brazil by the Argentinian party, Einstein brought back to life an idea he had previously commented in 1912 to Professor Julius Maurer, a colleague at Zurich. The idea was the possibility of measuring the displacement of stars near the uneclipsed Sun. In Maurer's opinion the idea was unpromising but at that time (1912) he suggested Einstein to consult George Ellery Hale, director of Mount Wilson Observatory. So in an effort to find a viable alternative to circumvent the problematic eclipse-hunting expeditions, Einstein finally did write to Hale a letter dated 14 October 1913. In this letter Einstein asks Hale: "*how close to the Sun fixed stars could be seen in daylight with the strongest magnification*". Hale answered that: "*there is no possibility of detecting the effect in full sunlight*" but he added: "*the eclipse method appears to be very promising [. . . ] and the use of photography would allow a large number of stars to be measured. I strongly recommend that plan*" [22]. Hale was right when he commented the null possibility of detecting the effect until observations started being made at radio frequencies. However, today geodetic Very Long Baseline Interferometry (VLBI) is capable of measuring the deflection of the light from distant radio sources anytime and across the whole sky [25].

Other eclipses were to follow, together with its respective attempts to validate Einstein's prediction. By early 1914 Freundlich planned to mount an expedition to Crimea, in the south of the Russian Empire, to verify Einstein's prediction during a next coming solar eclipse to occur on August 21 of that same year. He had previously discussed these plans with Einstein. However, the director of Berlin's observatory, where Freundlich labored, refused to cover expedition expenses. Nevertheless, Einstein obtained funding for the expedition, from the German industrial conglomerate Krupp, so plans to go to Crimea were not altered [26].

When the time arrived, Freundlich led the expedition to southern Russia. Essential parts to furnish some telescopes were lent to Freundlich by his Argentine colleagues whom he was going to

meet later in the port of Feodosiya in the Crimea, for joint observations [24]. The Argentinian party embarked to Russia some instruments they had previously employed in their previous 1912 failed attempt. A separate expedition led by William W. Campbell of Lick Observatory went to a different location near Kiev, Ukraine for the same purpose [27]. But World War I broke out while the three groups were already there and as a consequence, the German team became suddenly a war enemy. Freundlich was jailed in Russia and his instruments were confiscated, so he was helpless to make any observation. On the other hand, the Argentinian group in turn was also unable to make observations, in part as their own instruments arrived late and in part because they had loaned some of the seized instruments to Freundlich [24]. William Wallace Campbell, from neutral America, was permitted to continue with his plans, but a cloud cover hid the eclipse [26]. The members of both, the American and Argentinian expeditions hastily returned to their observatories, and their instruments had to be left in Russia under the custody of Pulkovo observatory for the extent of the war, a circumstance that hampered a subsequent opportunity to observe the next 1916 eclipse in Venezuela. On the other hand, Freundlich and his junior colleagues were jailed in Odessa for few months and were later released during an exchange of German prisoners for Russians caught in Germany when hostilities broke out [22].

## 6. The 1919 Eclipse. A Scientific Milestone

In 1915 Einstein had completed his GR. His next step was to apply the full field equations to solve pending physics problems. In particular, he recalculated the value of the deflection angle getting this time 1.75″ value which doubles his previous result, the factor of two arising because of both time and space contributions were now considered, whereas the first calculation took into account only the temporal curvature contribution [28].

To test this further prediction, new eclipse expeditions were to come. For the 1916 eclipse the Argentine Observatory managed to send a group to Venezuela, but the instruments they carried were not adequate to measure the sought light deflection [29]. Their appropriate instruments were still retained and stored in some warehouse in Russia.

Then again, a privileged opportunity to observe a total eclipse occurred in 8 June 1918, as the path of totality passed through United States. Yet, the Lick observatory did not have their apparatuses to photograph it even though international war in Russia was over as a separate peace treaty (Treaty of Brest-Litovsk) was signed between the new Bolshevik government of Soviet Russia and Germany on March 1918. By the end of April 1918 Lick's instruments set sail from Vladivostok Siberia for Kobe, Japan. The instruments remained at Kobe for a while as the final armistice on November 1918 put end to the First World War. Fearing that their instruments would not arrive on time, in an effort not to miss the opportunity to observe the eclipse, Lick's astronomers borrowed two lenses from the Chabot Observatory of Oakland, California, and other supplies from the Students' Observatory and the Department of Physics at Berkeley and equipped the aforementioned lenses with plate holders and cameras [22]. The improvised apparatuses were taken to the eclipse station at Goldendale, Washington St. On this occasion Lick's astronomers did manage to take photographic plates but, for some unclear reason results were never published. However, a preliminary oral report on the 1919 eclipse was delivered to the Royal Astronomical Society by Campbell, then Director of the Lick Observatory. He reported that in his opinion their 1918 eclipse measurements definitely ruled out the value for the deflection predicted by Einstein's theory [30].

The next eclipse in 1919, was to become the most famous in the history of 20th-century astronomy. A couple of years before, the Astronomer Royal Sir Frank Watson Dyson published a paper in the Monthly Notices of the Royal Astronomical Society pointing out that the total solar eclipse of 29 May 1919 would be particularly advantageous for investigating the deflection of light by the Sun, as the totality of the eclipse was going be unusually long, lasting about 6 min. In addition, he remarked that the Sun would have as background the Hyades star cluster, rich in bright stars and thus suitable for deflection of light measurements. Dyson exhorted astronomers to take advantage of such excellent

and exceptional opportunity [31]. However, there was an inconvenience, the path of totality ran across the Atlantic from Brazil to West Africa (Figure 2). This brought about limited options for the selection of observation sites.

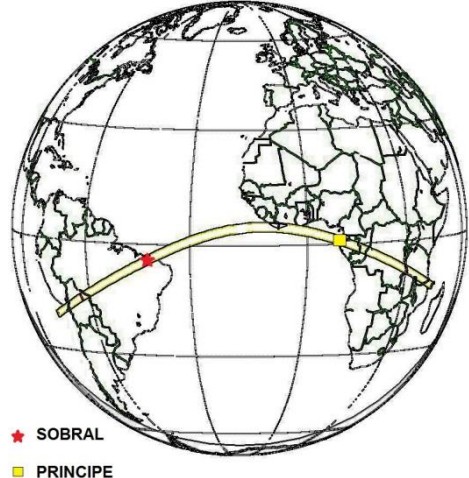

**Figure 2.** Path of the total solar eclipse of 29 May 1919. Locations of Sobral and Principe are shown.

In Britain there was a group devoted since 1893 to eclipse expeditions, the Joint Permanent Eclipse Committee (JPEC) [32]. After discussing on where to make observations, JPEC decided to send two expeditions: one from the Cambridge observatory and a second one from Greenwich, both under the overall direction of the Astronomer Royal, Sir Frank Dyson. The Cambridge party to be led by Arthur Stanley Eddington, to the island of Principe off the west coast of Africa. This party would take Edwin Turner Cottingham a clockmaker as assistant technician in charge of all the clocks and coelostats used to track the movement of the Sun during the eclipse. The Greenwich group to be directed by Andrew Claude de la Cherois Crommelin assistant astronomer at the Royal Greenwich Observatory and Charles Rundle Davidson, to Sobral in northern Brazil (Figure 2). The sole purpose of both expeditions was to observe a total solar eclipse that took place on 29 May 1919 to test Einstein's prediction, as already mentioned.

Soon Dyson, in charge of organizing both expeditions, faced several problems that he had to solve if he wanted the expeditions to succeed. To start with, in 1917 the First World War was taking place (July 1914 to 11 November 1918) and obviously sponsoring scientific expeditions was not a priority of the British government. Another problem was that Eddington, who was to lead one of the two parties, was the age of military recruitment and because of his condition as a Quaker and pacifist, he was a conscientious objector. The problem was obviously not his refusal to contribute to the war effort but what was in store for him the rest of the war. In 1917 recruitment became mandatory in the UK and his fate was then to be detained with other Quaker friends and forced to peel potatoes in Northern Ireland. Dyson was aware of the importance of Eddington involvement as the leading member of one expedition group. Eddington was an eclipse veteran. He led an eclipse expedition to Brazil in 1912. Besides he had worked on parallax analysis of asteroids on photographic plates. For that task Eddington had developed a novel statistical method based on the apparent drift of two background stars, winning him the Smith's Prize in 1907. Therefore, these credentials made Eddington the ideal candidate to lead one of the 1919 expeditions to measure the predicted bending of light. Given these facts, Dyson negotiated with the Home Office to make Eddington an exception, under the condition that the war been over before the eclipse happening, otherwise he ought to enlist.

In addition, a shortage of skilled labor made matters worse for Dyson and organizers. The importance of the future eclipse merited the use and construction of dedicated equipment, which required highly specialized technicians who were currently engaged in the war effort or were at the front. Even if Dyson and colleagues were able to gather the ideal instruments for a successful

expedition, these would have to be transported by ship through the dangerous waters of the English Channel, infested by U-boats and German warships. Almost nothing could be done until war ended. Finally, armistice was signed in November 1918 leaving less than six months to complete all arrangements.

The funding granted by the government was 100 pounds for adapting telescopes and 1000 pounds for various expenses. Finally, after a hectic time, both expeditions succeeded to deploy their instruments at their observation sites. During the observation of the eclipse both expeditions suffered setbacks either due to bad weather or instrumental problems. However, they managed to take some pictures of the event. Upon their return to England they brought their photographic plates with them. A subsequent analysis of the plates was coordinated by Dyson and the decision of which plaques should be discarded as unreliable data and which should be considered was taken jointly by him in consultation with Davidson, and not by Eddington [33]. This is important to remark as Eddington was an enthusiast of Einstein's General Relativity while Dyson was a sceptic.

The outcomes of both expeditions were made known at a special joint meeting of the Royal Astronomical Society and the Royal Society of London, convened on 6 November 1919. The announced results were roughly consistent with Einstein's prediction of General Relativity and firmly ruled out the only other theoretically predicted value, the so-called "Newtonian" deflection. The result from Sobral, gave a deflection of 1.98″ ± 0.16″ while Principe result was a less convincing angle of 1.61″ ± 0.40″. Both were within two standard deviations of the Einstein value of 1.75″ and more than two standard deviations away from either zero or the Newtonian value of 0.85″ [1].

Details of how the mentioned values were obtained were given during that meeting. For the value of Sobral, only the measurements from the smallest telescope of the two instruments employed there were used. These were consistent with Einstein's prediction. All measurements taken with the larger astrographic telescope were left out on the grounds of having focusing problems with that telescope, at the time of the eclipse. A decision taken jointly by Dyson and Crommelin. However, these neglected results yield a value for the deflection of 0.93″ seconds of arc, value which is very close to the Newtonian prediction [34]. It is important to remark that in 1979 a re-analysis of the original and neglected Sobral photographic plates, with modern measuring equipment and contemporary software yielded a value of 1.87″ ± 0.13″ a result which is just within one standard deviation of the predicted value, vindicating the 1919 conclusions [35].

The audience reaction at this special meeting was ambivalent. Some questioned the reliability of the measurements and showed skepticism as in their view few stars were used to determine the deflection angle (Figure 3). Others suspected Eddington of cooking the books. But, we have commented above that in fact was Dyson, not Eddington, who gave his approval to selecting which data were good enough to consider.

The accuracy of the 1919 results was indeed poor but adequate to persuade the mainstream of contemporary astronomers. The reported results were hailed at the time as a conclusive proof of GR over the Newtonian model. The latter conclusion was published in newspapers all over the world as a major story. As a result, Einstein was fame catapulted.

However, in the conclusion of the celebrated paper read at the November 1919 joint meeting of the Royal Astronomical Society and the Royal Society of London, the authors advised "*. . . the observation is of such interest that it will probably be considered desirable to repeat it at future eclipses*" [1]. For very detailed narratives of the eclipse expeditions and events that followed see also [36–38].

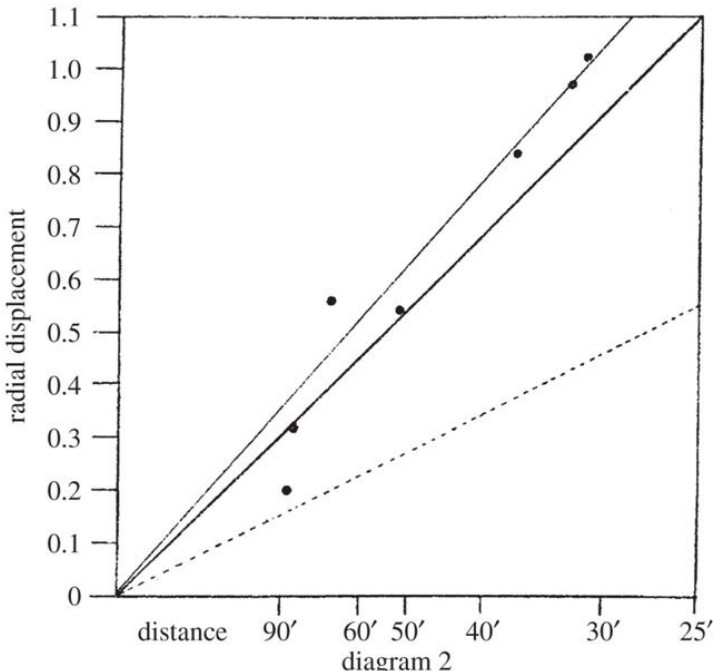

**Figure 3.** The radial deflections of the positions of seven stars observed by the 4-inch telescope at Sobral as a function of distance from the center of the Sun. The scale on the abscissa is the inverse of the distance from the centre of the Sun. The dotted line shows the Newtonian prediction and the central heavy solid line shows the expectation of the General Theory of Relativity. The upper light solid line shows a best-fit to the deflection of the seven stars by the Sun. Image taken from [1].

## 7. Half Century (1922–1972) of Eclipse Surveillances

The shocking result obtained by the British expeditions of 1919 would definitely change our image of the universe. This awakened worldwide attention to ensure that the measurements had indeed been correct and that Einstein was consequently right.

An opportunity to verify Einstein's GR came soon. This came in 1922 when an eclipse was observed at the Cordillo Downs sheep farm in South Australia, near the Queensland border [39]. This time copious data were obtained by Lick's observatory team showing the displacements of many more stars that those measured in 1919 (Figure 4), but then again the uncertainty remained stubbornly about 0.2–0.3 arcsec.

Eclipse deflection measurements by optical methods have been repeated several times throughout half a century since the 1919 eclipse. They all shared the common feature of considerable uncertainties in their obtained values.

The idea of optical method is simple: to photograph the sky fragment around the Sun during eclipse totality and to compare the arrangement of the same stars without the Sun. Figure 4 shows changes in star positions recorded by Lick's team in Australia during the eclipse of 21 September 1922 [39]. The eclipsed Sun is represented by the circle in the center of the diagram, surrounded by a representation of the coronal light. Stars too close to the coronal light cannot be used. The recorded displacements of other stars are represented by lines. The displacements are statistically processed sometimes with preferable weights assigned to different stars.

Measuring the deflected light rays in this manner, using optical telescopes, continued into the 1970s but never increased much in accuracy [40]. This is due to the difficulties in observing stars through our atmosphere. As we have already pointed, considerable uncertainty remained in these measurements for almost fifty years, until observations started being made at radio frequencies by interferometric methods. Results from several major expeditions are shown in Figure 5, including modern accounts on this measurement [41].

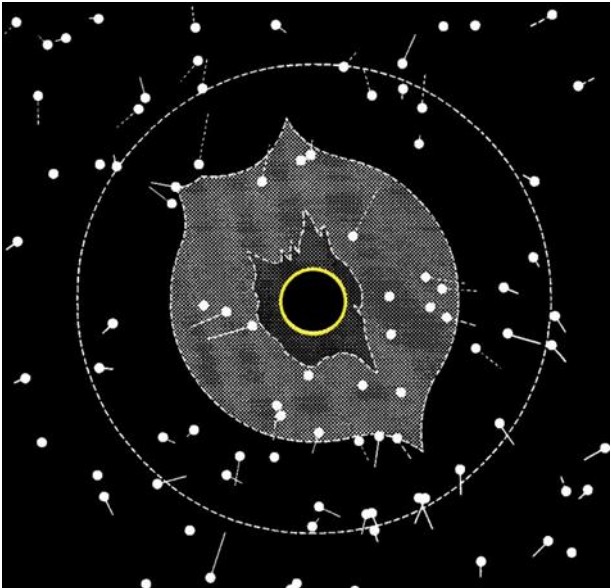

**Figure 4.** Changes in star positions recorded during the eclipse of 1922 and published in Campbell & Table 1923. Figure modified after ref. [39].

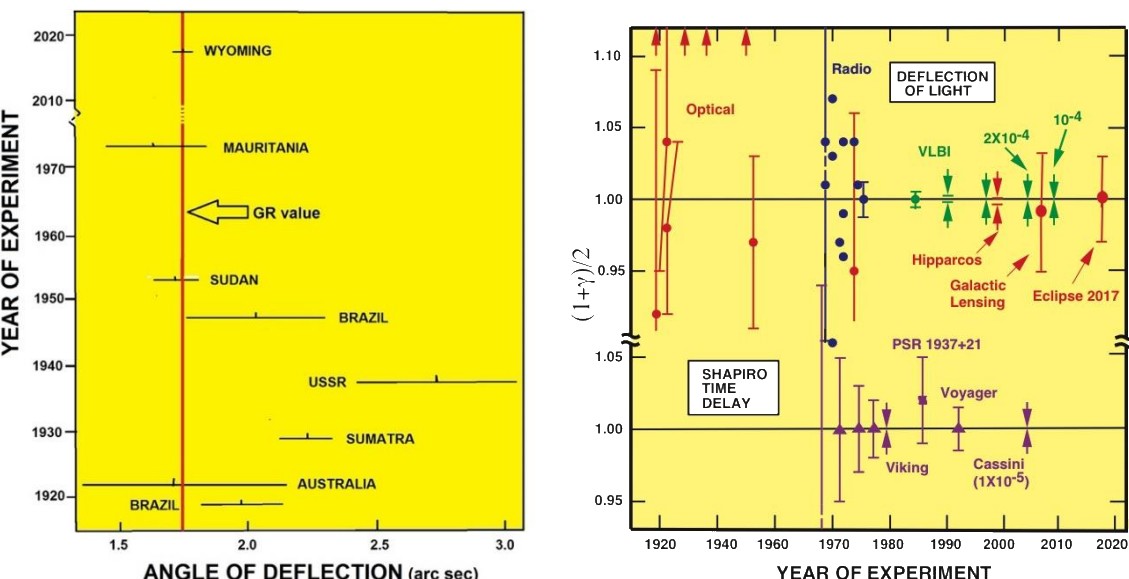

**Figure 5.** Left panel: Measurements of the angle of light deflection by the Sun. The red dashed line indicates the value given by GR. Right panel: Measurements showing how well GR is confirmed and providing additional constraints of the coefficient $(1 + \gamma)/2$ from light deflection and time delay observations. For GR $\gamma = 1$. In modified gravity the bending angle by the Sun would be $\delta\theta \approx 1.7505''$ $(1 + \gamma)/2$. The arrows at the top denote anomalously large values from early eclipse expeditions. The Shapiro time-delay measurements using the Cassini spacecraft yielded an agreement with GR to $10^{-3}$ percent, and VLBI light deflection measurements have reached 0.01 percent. Hipparcos denotes the optical astrometry satellite, which reached 0.1 percent. Credit right panel: C. M. Will, updated figure from the one in [41].

## 8. Gravitational Lensing Genesis 1912

In March 1912 an astronomical event of some relevance showed up in the skies. This was the appearance of Nova star (Geminorum 1912 (DN Gem)) [42]. This Nova event happened a month before Einstein had his first meeting with Erwin Finlay Freundlich in Berlin, during the week of 15–22 April 1912. It is almost certain that this prominent event was conversed in the course of Einstein encounter

with Freundlich. Prof. Tilman Sauer sustains that the Nova episode probably motivated Einstein to produce his first gravitational lensing equations [43]. The set of equations is included in an Einstein's scratch notebook that was written between, 1910 and 1914 [44]. These equations are the same than those that Einstein would later publish by 1936. These latter equations were published at the insistence of a Czech electrical engineer as we shall explain in a following section.

According to the opinion of Prof. Sauer, Einstein's motivation to produce the equations was his "idea of explaining the phenomenon of new stars by gravitational lensing" [45]. Einstein's idea is now known as microlensing induced variability, which is the effect that might be produced when a distant star is brightened by a near star due to its lens effect, see details in Section 18. However, Einstein abandoned this thought by 1915. In a letter to his friend Heinrich Zangger, dated 8 or 15 October 1915, Einstein made a side remark that "new stars" have nothing to do with the lensing effect [45]. It is worth mentioning that the idea of a star brightening another star was independently resurrected by Eddington in his book "*Space Time and Gravitation*", published in 1920, where it can be read [46]:

> "*If two independent stars are seen in the same line of vision within about 1″, one being a great distance behind the other. . . It would seem that we ought to see the more distant star not only by the direct ray, which would be practically undisturbed, but also by a ray passing round the other side of the nearer star and bent by it to the necessary extent. The second image would, of course, be indistinguishable from that of the nearer star; but it would give it additional brightness*".

From these comments it is clear that Eddington was not claiming the possibility of observing a double image of a single star, but only two stars, one brightened by the other. Sometimes it is asserted in the literature that Eddington proposed the idea of multiple images, but that does not seem the case.

As a matter of fact, it seems that the first person to raise that possibility was Orest D. Chwolson of Pulkovo observatory in Petrograd. In 1924 he published a short note under the title, "On a possible form of fictitious double stars", where he considered the idea of a "fictitious" double star and the mirror-reversed nature of the secondary image. In addition, he established the conditions under which one can observe what is now known as Einstein's ring [47]. This occurs when the source, lens and observer are all aligned, resulting in a circular image around the lensing object. At first glance it would seem that Chwolson's short note came "out of the blue" but it is almost certain that Pulkovo astronomers were acquainted in gravitational light–deflection. The fact that the instruments of Campbell's eclipse expedition of 1914 were temporarily stored in Pulkovo observatory might have contributed to call their attention.

To conclude this section, it is thought-provoking to indicate that in the same page that Chwolson's note was published and right below his note, there is a comment written by Einstein answering a remark by W. Anderson on a different issue not related to light bending [48]. One may question if Einstein ever read Chwolson's note.

## 9. Czech Mates

In 1920 Arthur Eddington wrote a book entitled "*Space, Time and Gravitation: An Outline of the General Relativity Theory*" [46]. The book explained Einstein's GR to the general public. This publication immediately became a best-seller among educated readership. In it, to explain Einstein's 1916 eclipse prediction, Eddington pointed out the analogy between gravitational deflection and optical refraction. In its pages one can read that: "*Any problem on the paths of rays near the Sun can now be solved by the methods of geometrical optics applied to the equivalent refracting medium*". This phrase could be taken as a call or invitation to someone with knowledge in optics to initiate what could be a new field of research. But for years no one answered that call.

Around the beginning of 1930's Frantiŝek Link, a young Czech meteorologist and astronomer, was engaged in the study of layers of the Earth's atmosphere using photometric surveys during a lunar eclipse. Link's idea was to investigate the density of atmospheric layers by measuring the intensity of light rays reflected on the eclipsed face of the moon. Before impinging on the moon's surface, those

rays tangentially cross the Earth's limb and therefore are refracted by our atmosphere toward the eclipsed face of the moon [49]. As it is known, according to GR, a gravitational field bends light in much the same way as atmospheric air layers (with vertical density gradient) curve the trajectory of a propagating light ray. These formal similitudes between gravitational bending and geometric (atmospheric) refraction, as Eddington had already pointed years before, lead Link to publish in 1936 a breakthrough paper on gravitational optical lensing where he computes not only the position of the images, but also their brightness, and considers both visible and invisible stars as possible sources, remarking that the amplification produced by the lens might, in particular circumstances, turn the image of a faint star and otherwise invisible, into a visible one. This work was published in French on 16 March 1936 [50]. Unfortunately, Link's paper appears to have passed totally unnoticed among astronomers. During the 1936 summer, Link further develops his calculations and produced a second paper where he included finite size effects of sources and lenses. He remarked that the invariance of surface brightness of a source may produce distorted image shapes such as: arclets, lentils, and rings [51], see Figure 6.

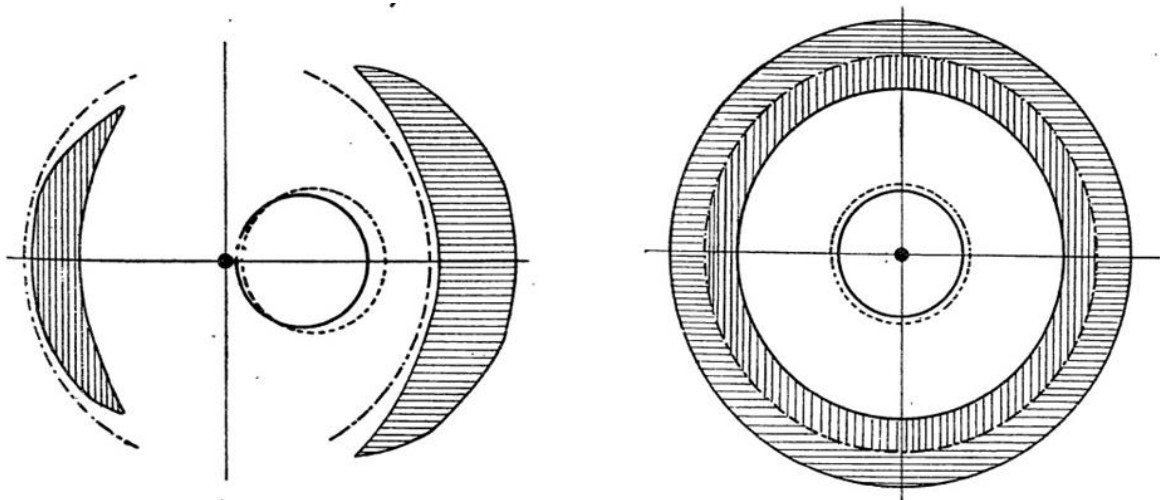

**Figure 6.** Stages of stellar body (large circle) eclipsing a source (small dot). Shadow areas represent distorted images as seeing them from Earth. Picture taken from [51], source: http://articles.adsabs. harvard.edu/pdf/1937BuAst..10...73L.

Link's remarkable second paper (1937) concludes encouraging readers to "*systematically search for such phenomena in all domains of stellar astronomy*" [51].

Meanwhile in the Spring of 1936, when Einstein was living in Princeton—having emigrated to the US three years earlier—he received a visit from Robert (Rudy) Welt Mandl, a Czech electrical engineer. Mr. Mandl was an immigrant with limited economic resources because at that time he made his living as a dishwasher in a restaurant in Washington, D.C. and also he was making an extra income by selling hand painted eggshells with intricate geometric designs [52]. Mandl had come up with some far-reaching ideas and he insisted Einstein that he should calculate the effect of gravitational focusing of light during stellar eclipses.

Einstein had already made such calculations on gravitational lensing in a scratch notebook dated to the spring of 1912, apparently motivated by the Nova Geminorum event of 1912, as we have already mentioned in a previous section. At first Einstein was reluctant to concede Mandel's wishes as he was convinced that the phenomenon was unobservable. However, during the next months, Mandl exerted so much pressure that Einstein finally sent a note to *Science* entitled "Lens-Like Action of a Star by the Deviation of Light in the Gravitational Field" [53]. The note contains the final formulas without any derivations and was published on 4 December 1936. The first paragraph of Einstein's note reads: "*Some time ago, R. W. Mandl paid me a visit and asked me to publish the results of a little calculation,*

*which I had made at his request. This note complies with his wish*" and Einstein concludes his note with a very skeptical comment: "*Of course, there is no hope of observing this phenomenon directly*". Shortly after, Einstein vented his pessimism openly. In a letter to the editor of *Science*, James Cattel, Einstein wrote, "*Let me also thank you for your cooperation with the little publication, which Mr. Mandl squeezed out of me. It is of little value, but it makes the poor guy happy*" [54].

It is interesting to mention that Einstein, in the title of his 1936 paper accurately used the phrase "lens like" instead of "lens", because there are two main differences between a lens and what today we call a gravitational lens. The former has defined focal point whereas the latter has not necessarily a unique focal point (Figure 7). The first person to point this out was Oliver. J Lodge in 1919, who remarked that it is "*not permissible to say that the solar gravitational field acts like a lens, for it has no focal length*" [55]. The second difference between an ordinary lens and a gravitational one is that the latter is always achromatic. Einstein's 1936 paper was perhaps unintentionally responsible for introducing the term "lens" to the newborn field as no previous paper seems to have used the word "lens" to refer to the phenomena before (the term was not "permissible" to Lodge). Though there are these differences, in practice one finds configurations that possess a well-defined focal plane, e.g., when one approximates a lens distribution as a sheet of uniform surface density, and in this case, the analogy to a lens is well approximated, except only for the spatial dependence of the lens refractive index.

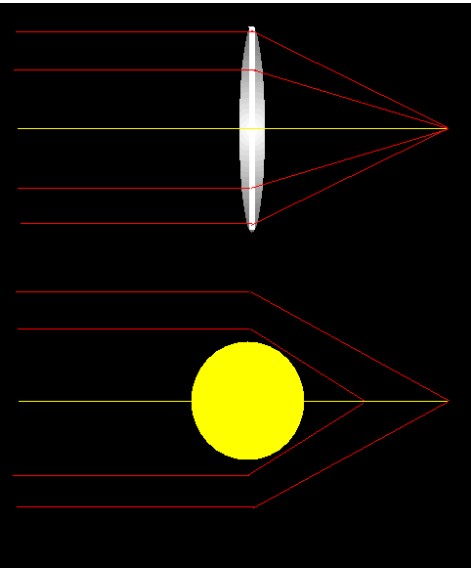

**Figure 7.** Ordinary convex glass lens and the Sun. The Sun produces a maximum deflection of light that passes closest to its limb, and a lesser deflection of light that travels furthest from its center. Consequently, the Sun's gravitational lens has no single focal point as the convex glass lens, but a focal line.

To end up this section, a natural question arises. Was there a link between Link and Mandel? On 16 March 1961, F. Link delivered a talk entitled "Einstein's light-deflection in modern astronomy" at the Paris Astrophysics Institute. In it he asserts "*It may be useful to add that we did not know our fellow countryman R. W. Mandl*" [56].

It is interesting to add that Mr. Mandl was a very determined person as he did not only pay a visit to Einstein, he had also contacted V.K. Zworykin urging him to investigate gravitational lensing. Zworykin was a famous electrical engineer at that time and one of the inventors of television. Later Zworykin met the Caltech astrophysicist Fritz Zwicky and mentioned him: "*the possibility of an image formation through the action of gravitational fields*" [57].

## 10. The Quarter Century Interregnum (1937–1963)

Einstein's publication together with Zworykin's comment, provoked Zwicky to investigate the subject. In 1937 Zwicky recognized, as Einstein did, that lensing between stars might be very difficult to observe, but he realized that this might not be the case for galaxies. He wrote: *"The problem in question, however, takes on a radically different aspect, if, instead of in terms of stars we think in terms of Extragalactic Nebulae [Galaxies]"* [54]. He was able to arrive at this conclusion owing to his idea that galaxies contained far more mass than what is visible. In fact, by 1933 Fritz Zwicky studied the motion of galaxies within the Coma Cluster. Using the virial theorem Zwicky calculated a cluster mass more than 400 times larger than that previously estimated from observations of luminous matter. Zwicky then concluded that if these measurements of luminous matter held true *"dark matter is present in much greater amount than luminous matter"* [58]. This is the first reckoned paper arguing the need of dark matter; there was however another astronomer, the Dutch Jan Hendrick Oort, who a year before also proposed the existence of dark matter but at galactic (not cluster) scales, arguing however that it could be due to hidden stars [59].

In a first paper published on the topic, Zwicky made some calculations that showed *"that extragalactic nebulae [galaxies] offer a much better chance than stars for the observation of gravitational lens effects"* [58]. In a second publication, Zwicky added that observations of light deflection might provide *"the most direct determination of nebular [galactic] masses"* [60].

However, Zwicky's pair of papers didn't stir up interest at that time among colleagues. After Zwicky's 1937 papers an interregnum of nearly a quarter of a century (1937–1963) followed. Interest in lensing entered into a period of stagnation.

Yet it must be mentioned that there were two isolated responses to Einstein paper. The first was a reaction to Einstein's 1937 paper, by Henry Norris Russell who wrote a popular article, in which he considers a hypothetical situation in which an imaginary spectator situated on a planet orbiting Sirius B observes multiple images, arcs, and amplification effects, during a Sirius eclipse [61]. The second reaction was that to the 1924 short note by Orest Chwolson's from another Pulkovo's astronomer by the name of Gavil A. Tikhov. This 1937 work dealt with simple star–star lensing and image amplification extending Einstein's calculation on the intensities of lensed rays for a general case [62].

## 11. Theoretical Renaissance of Lensing (1963–1979)

In astronomy usually, observations come first and explanation later but not always. That is the case of gravitational lensing, as the basic physics of gravitational lenses was understood well before the first example emerged, actually discovered in 1979.

As already mentioned, the subject apparently fell in a latency period until the beginning of the sixties. New ideas of lensing applications were then suggested in the years 1964–1965, during this period, Yu Klimov, Sidney Liebes Jr., and Sjur Refsdal independently revived interest in the theory of gravitational lensing.

Liebes first version of his paper was a post-dead-line presentation at the August 1963 APS meeting. There he was the first to make a careful estimate on the frequencies at which events might occur, in particular on stellar lenses of various types. Later, in an extended version of his manuscript he suggested that stars might be surrounded by radial spikes of concentrated light intensity [63]. Klimov considered lensing by galaxies [64].

In 1964 Sjur Refsdal wrote a pair of fundamental papers, both communicated to the Royal Astronomical Society by Hermann Bondi. In the first he described the properties of a point-mass gravitational lens, simplifying calculations previously made by Gavil A. Tikhov in 1937, arguing that geometrical optics could be used for gravitational lensing [65]. In the second paper he demonstrated that the Hubble parameter ($H_0$) and the mass of a galaxy can be expressed in terms of time delay, redshifts of both the lens and the source, and the angular separation of the lensed images (see Section 17) [66]. This method was thought to be applied to a supernova lying far behind and close to the line of sight through a distant galaxy, however, it was only until much later measured [67]. In that paper Refsdal

also called the attention on the potential importance of quasars in gravitational lenses. It is pertinent to recall that some time before (1963) the first quasar, 3C 273, a "quasi-stellar" compact, very luminous and distant source, was identified by Maarten Schmidt [68]. It is worth mentioning that its identified cosmic origin sparked attention in lensing. Then, interest arose to study events that would lead to quasar identification [69].

In August 1965 in the 119th meeting of the AAS at Ann Arbor, J. M. Barnothy mentioned the influence of lensing on the quasar phenomenon. He asserted that source counts of QSO's and their short-time brightness variations can be affected by the lensing action of foreground galaxies [70].

Two more articles were published by Refsdal in 1968 where he suggested the use of lensing for testing cosmological theories such as the then popular steady state theory, and theories based on GR [71]. The second paper dealt with the conditions to determine the mass and distance of a star which acts as a gravitational lens, if the lens effect can be observed from the Earth and from at least one distant space observatory [72].

By that period, it was already realized that light propagation in a real universe whose mass distribution was not homogeneous should differ from a homogeneous universe because there are regions of space with greater mass density than others and consequently with different lensing effects. This fact should then be considered. A third paper was published by Refsdal at the beginning of 1970 on the influence of lensing on the apparent luminosity of very distant light sources in static and flat universes with inhomogeneous mass distributions [73]. In this article he studied the validity and limitations of this model and possible extensions to expanding and curved universes.

Throughout the seventies, theoretical work continued but without any systematic observational search. In 1971 N. Sanitt published a work that regards galaxies (extended masses) as lenses and considered its influence of lensing (amplification bias effect) on source counts of QSOs [74]. In 1973 motivated by the conviction that some galaxies are thought to contain a spheroidal mass component Bourassa and co-workers studied the properties of spheroidal lenses such as intensification, distortion, and orientation of images around such galaxies [75].

## 12. Two Birds with One Stone, Q0957+561 A and B

In the mid-1970s the interest in discovering new quasars resurfaced. Most of the few quasars known at that time emitted powerful energy at radio wavelengths, so a usual detection procedure involved locating the approximate positions of candidates through the use of radio telescopes. Then, focusing optical telescopes to said candidates to find indicative signature of distant quasars, which in this case their spectra should show more intense blue light than red compared with an average star in our own galaxy.

At that time radio astronomers were starting to use pairs of radio telescopes as interferometers to obtain much more accurate positions of celestial sources of radio waves. At Jodrell Bank Observatory in England, there were two radio telescopes, the Mark IA telescope of 76-m diameter, with the smaller Mark II, of 25-m diameter; together these provided positions with errors as little as 2 s of arc for unresolved sources. Around the end of 1973, a source coded by 0957+561, was located. This was just one of 800 sources found by a team headed by Prof. Dennis Walsh of the University of Manchester [76].

Walsh then embarked on a program of optical confirmation of the quasar candidates from this survey with a number of collaborators. By March 1979 Walsh and Robert F. Carswell were doing routine optical spectroscopy of quasar candidates at Kitt Peak, on the 84" telescope. Near the radio position of one, the aforementioned 0957+561, there were two blue objects which were candidate quasars. The spectrum of the first of the two they chose to look at confirmed it as a quasar, and then they took a spectrum of the second. Its spectrum appeared to be identical. So much so that in words of Bob Carswell, "...*we wondered if the telescope operator had again pointed the telescope at the first object by mistake!*" [77].

Walsh and Carswell realized the importance of their discovery but neither of them knew, at that stage what that similarity of spectra meant. The next day Carswell called Ray J. Weymann to discuss

what they had found. By a lucky coincidence, Weymann was scheduled on the Steward Observatory 90″ telescope that very same night and so they decided that higher resolution spectra should be obtained at the 90″ to allow a more detailed comparison. The repeated observations of both objects confirmed their essentially identical nature. This measurement strongly reinforced the idea that the light was originally coming from a single object and that they were observing a double image produced by a gravitational lens [77].

The results of these observations and this proposed explanation were soon published by Dennis Walsh, Robert F. Carswell, and Ray J. Weymann [78]. They reported the double quasar Q0957+561 as two quasar images with the same color, redshift (1.41) and spectra, separated by only 6.1 arc-seconds, produced by a gravitational lens. Gravitational lensing had become reality. Figure 8 shows the double quasar Q0957+561. However, for a time some radio astronomers disputed this interpretation, but as we shall see next, further investigations confirmed it.

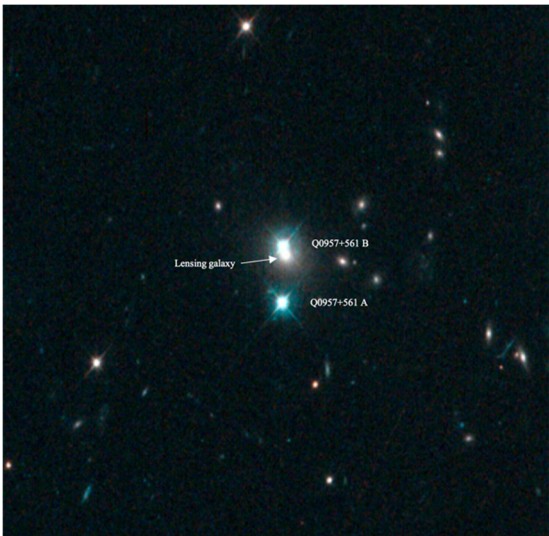

**Figure 8.** Components of double quasar QSO 0957+561 A and B) that are separated by 0.6″. Image credit: NASA/ESA Hubble Space Telescope/G. Rhee, processed by W. Keel.

The year 1979 also marked two important technical developments in astronomy: the first CCD detectors replaced photographic plates, thus providing much higher sensitivity, dynamic range and linearity, and the very large array (VLA), a radio interferometer providing radio images of subarcsecond image quality, went into operation. With the VLA it was soon demonstrated that both quasar images are compact radio sources, with similar radio spectra. Soon thereafter, a galaxy situated between the two quasar images was detected [79–81].

The galaxy has a redshift of 0.36 and it is the brightest galaxy in a small cluster. We now know that the cluster contributes its share to the large image separation in this system. Furthermore, the first very long baseline interferometry (VLBI) data of this system, known as QSO 0957+561, showed that both components have a core-jet structure with the symmetry expected for lensed images of a common source (see Figure 9). The great similarity of the two optical spectra (Figure 10) is another proof of the lensing nature of this system.

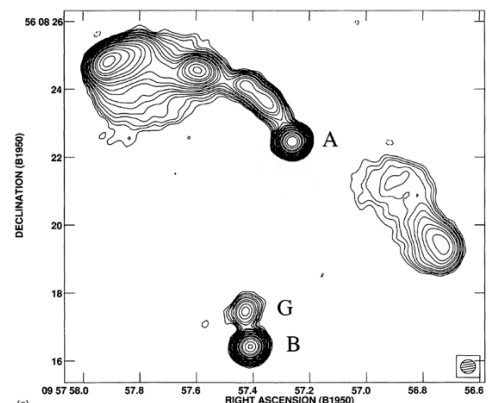
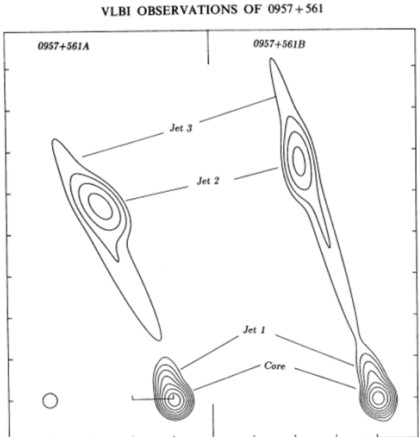

**Figure 9.** The left panel shows a 6-cm VLA map of the QSO 0957+561 [82] where besides the two QSO and the extended radio structure seen for image A, radio emission from the lens galaxy G is also visible. The milli-arcsecond structure of the two compact components A, B is shown in the lower right panel [83] where it is clearly seen that one VLBI jet is a linearly transformed version of the other, and they are mirror symmetric; this is predicted by any generic lens model which assigns opposite parity to the two images. Images credits: [80,82,83].

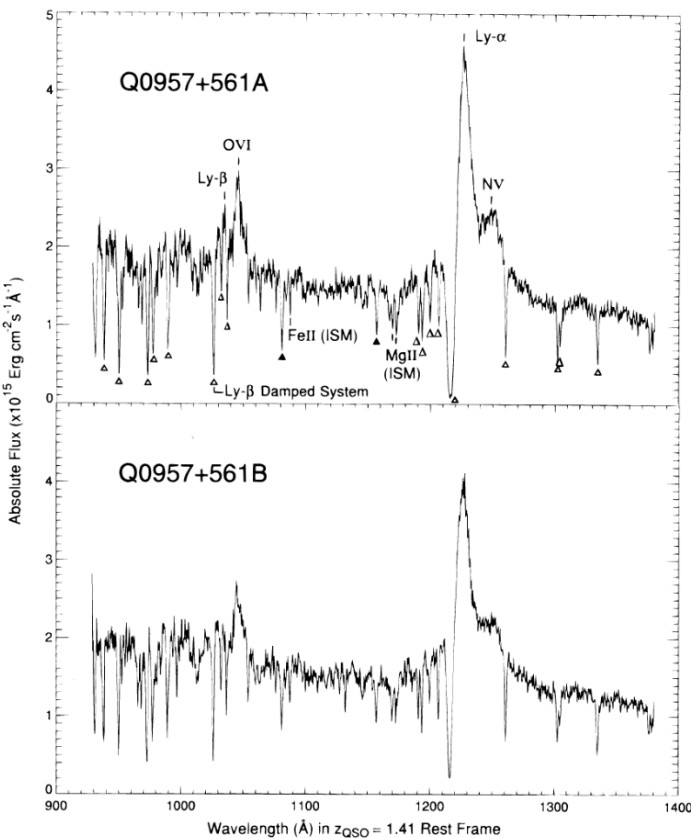

**Figure 10.** Spectra of the two images of the lens system QSO 0957+561, obtained with the Faint Object Spectrograph on board HST [84]. The strong similarities of the spectra, in particular the same line ratios and the identical redshift, verifies this system as a definite gravitational lens system. Image credit: [85].

## 13. Beyond Double Images (1979–1984)

The discovery of the double image of a lensed quasar QSO 0957+561 in 1979 did not seem to arouse immediate interest in organizing a systematic search for similar cases. In fact, the second

example of a lensed quasar was serendipitously discovered in 1980. This second sighting had its roots in 1976, when the American astronomer Richard F. Green of Hale Observatory undertook a program to register bright quasars in the Northern Hemisphere [85]. The method he used for classifying new quasars was by spectroscopic analysis of each one of the more than two thousand objects enlisted in his catalog. This task was going to take him and coworkers several years. By 1980 they unexpectedly discovered that three of the examined quasars had the same spectral content and were also very close to each other. They realized then that they had found a triple image of the same quasar. This fortuitous discovery marked the second encountered gravitational mirage [86]. Later on, in 1987 its lensing object was identified as a spiral galaxy [87]. A subsequent analysis using speckle interferometry revealed that gravitational lens produced an additional faint image. This latter study, with a better resolution, revealed that the gravitational lens actually produces four images and not two as originally supposed [88]. Third and fourth cases of gravitational lensing were also discovered by chance [89,90].

This series of serendipitous discoveries, in a span of five years (from 1979 to 1984) finally constituted the incentive to tie the interest of some towards running systematic searches. Soon afterwards a pioneer survey was carried on. Starting from a catalog of a few thousand radio sources, an American team from MIT, Caltech, and Princeton, selected those sources that presented on radio maps very good resolution and a multiple structure. Once pinpointed, they optically observed them carrying out a spectroscopic study. As result of their observations, they found a double image compatible to a gravitational lens system [91]. As an additional bonus of their publication, they suggested a search course of action to find new multiple imaged quasars.

Ever since mid-eighties, there has been a rapid and steady growth of research work in the field of gravitational lensing and this in turn has activated the growth of multiple and diverse areas of the field. These areas have grown similarly to the branches of a weeping willow, that is, there are many branches of equal importance with simultaneous parallel development. It is for this previous reason that from now on, we shall abandon the chronological approach and we will group the narrative of our saga in the subjects that next we will be treating. Most of the lensing effects whose theoretical bases were sketch out in the 60s and 70s have now been observed. Some of them have given rise to solid topics of research. In what follows we shall describe some of these branches. Our choice is mainly based on the use of gravitational lensing as an astronomical and astrophysical tool, apologizing for those omissions and oversights which undoubtedly have been made.

## 14. The Arrival of Charge Coupled Devices (1980's)

At a discovery rate of approximately of one lensed quasar per year, some astronomers of that time (mid 1980's) began to be aware that the gravitational lensing phenomena could be present in their daily observations. With the advent of new, better and more sensitive instruments the discovery rate of multiple image quasars began to increase promptly. In particular, the use of CCD's facilitated this task as its use became common [92]. As an example, in 1984 a sky survey found nine new "independent" quasars using photographic plates [93]. Four years later one of these quasars was subsequently reexamined by a different team, and the image was captured with a CCD [94]. This time the single-image quasar turned out to be a lensed object which consisted of four bright image components (Figure 11).

Even though the introduction of CCDs in optical observatories as well as new and better technologies in radio observatories, paved the way to the discovery of new multiple imaged quasars, the identification of lensed images presents problems as it is not always a straightforward task. Eventually binary quasars had been erroneously identified as lensed quasars, as the two quasars in a binary system may possess very similar characteristics. The most renowned case of misperception between binary and lensed quasar is that of QSO B2345+007A and QSO B2345+007B [95]. These two quasars have very similar optical images and spectra and were mistaken as two images of a lensed quasar, however further observations by the Chandra X-ray Telescope have shown that they belong to a binary quasar [96].

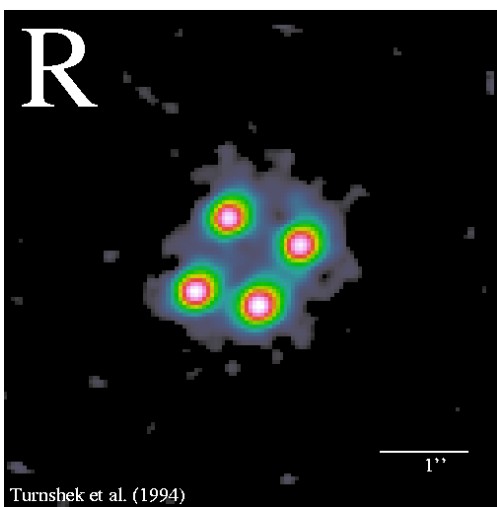

**Figure 11.** "Clover leaf" quasar H1413+117. The quasar was found in 1984 as a single object, then in 1988 it was discovered to be a lensed quasar split into four images. Image credit: NASA/STScI/D.Turnshek.

At the present time dozens of gravitational lens systems producing multiple images have been clearly identified. By late 2018, the CfA-Arizona Space Telescope Lens Survey (CASTLeS) had listed 82 verified gravitational lenses producing multiple images, 10 unconfirmed lenses and 8 possible cases [97].

Next we shall narrate a spectacular observation of lensed system that happened on January 1987 that puzzled some astronomers and caught the attention of the general public. This event certainly contributed to make awareness of the consequences of gravitational lensing.

## 15. Lensing, a Tool to Get Mass-Density Distributions

It was in 1986 during the 169th meeting of the American Astronomical Society that Roger Lynds of Kitt Peak and theorist Vahe Petrosian of Stanford University reported the existence of Giant Luminous Arcs in galaxy clusters [98].

While surveying clusters of galaxies for other purposes, Lynds observed two, possibly three examples of bright luminous arcs stretching between galaxies in three of the 58 clusters of galaxies he surveyed. The first arc was seen in the vicinity of the galaxy cluster Abell 370 (Figure 12). The other was near galaxy cluster 2242-02. The observed arcs were in excess of 100 kpc in length and luminosities roughly comparable to those of giant elliptical galaxies [98]. The discovery of such large "objects" was unprecedented as these arcs appeared to be the largest objects that had ever been observed, at that time, in the universe. This apparently outstanding discovery made news in the media [99,100].

At the time of the discovery, Lynds and Petrosian began to speculate what the arcs could be, as they appeared to be the largest objects that had been observed to that date in space. So the questions were: what they were made of? and how they got there? Initially they thought that the blueness of the arcs might indicate that the arcs were composed of young stars formed along an advancing shock front. Young stars are blue; mature stars tend to be yellow or white. Hence their first task was to obtain spectra to determine whether there were stars in the arcs, or whether arcs were simply composed of luminous gas.

Some months later in August 1986, a French team headed by Genevieve Soucail and colleagues of the Observatoire de Toulouse, gave an account of unreported observations of the same arcs they had made in 1985 [101]. In their paper they speculated that the arcs might be the result of galaxy/galaxy interactions or of star formation occurring from a cooling flow of the intra-galaxy cluster medium.

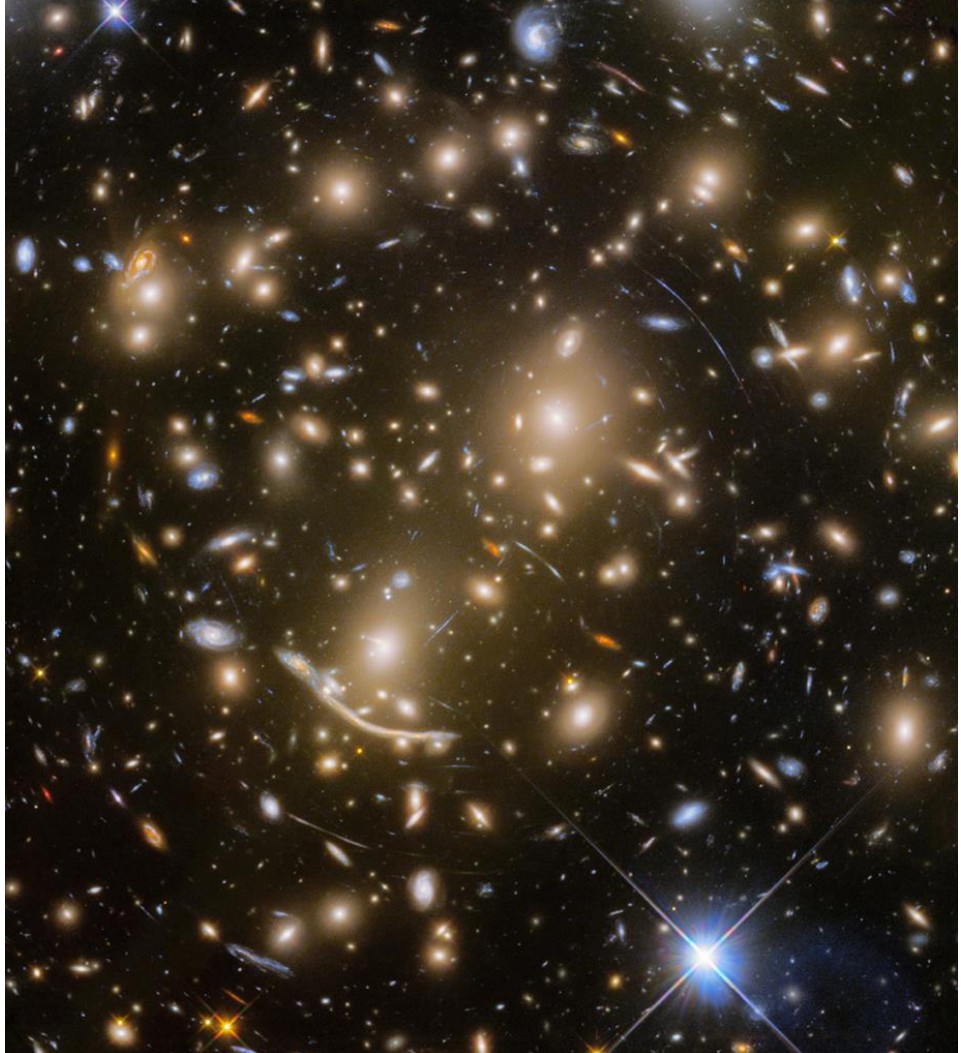

**Figure 12.** Bright luminous arcs stretching between galaxies in cluster Abell 370. Credit: NASA, ESA/Hubble, HST Frontier Fields.

It was few months after the report of Lynds and Petrosian observation, that Bohdan Paczynski proposed they could be gravitationally lensed images of galaxies [102]. He wrote: "it is possible that they are images of galaxies located far behind the clusters". In addition, Paczynski suggested a way to prove his hypothesis by measuring redshifts of the arcs and of neighboring galaxy clusters, if the value of the former is larger than that of the supposed lensing cluster, then that is "*an unambiguous test of the gravitational lens hypothesis*". Further investigations confirmed Paczynski's hypothesis that in effect, these arcs are images of a distant galaxy lensed by the cluster Abell 370 [103,104].

Without too much delay, the first models for the lens producing the enormous arcs appeared in the literature [103,105,106]. From the very beginning it was clear that these models required a large amount of dark matter in the clusters as these lensing effects could not be explained by accepted theories of gravity unless more matter was present than could be seen. This meant that the phenomenon could be used to study the distribution of matter (both light and dark) at cluster-scale, provided that accurate and reliable models for gravitational lenses were developed.

Over the years further detailed models of lenses reconstructions have been developed using different approaches and by the end of the last century it was widely recognized that gravitational lensing observations were unique and also ideal tools to probe the deflecting mass-density distribution in the universe, in particular it helped to infer its dark matter content and dark matter properties—see,

e.g., [107,108]. Today's models produce remarkably high-precision mass maps, particularly with imaging data from the Hubble Space Telescope (HST).

Another source of dark matter reconstruction comes from gravitation lensing of the Cosmic Microwave (CMB) photons that we will explain below in Section 18. Here we would like to emphasize that all sky maps of gravitational lensing are made via the anisotropy and polarization measurements of the Planck satellite and they reveal a map of the dark matter in the universe, as shown in Figures 13–15.

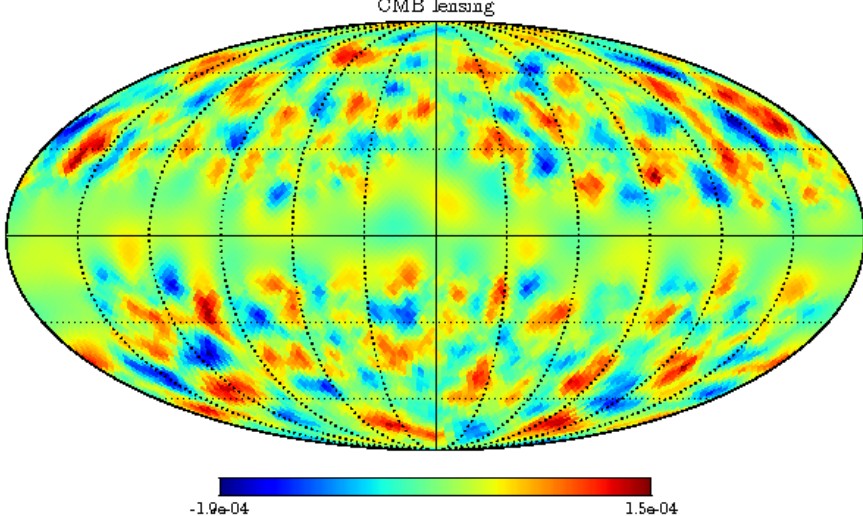

**Figure 13.** Sky map of the gravitational-lensing potential reconstructed from the CMB temperature fluctuations as measured by the Planck satellite (signal reduced along the galactic plane to avoid Figure 14. All-sky map of the CMB lensing potential constructed from the Planck 2015 data. Lighter regions correspond to integrated overdensities, darker to underdensities. The grey regions, where Galactic and extragalactic foregrounds are large, are masked in the analysis. Image credit: ESA and the Planck Collaboration.

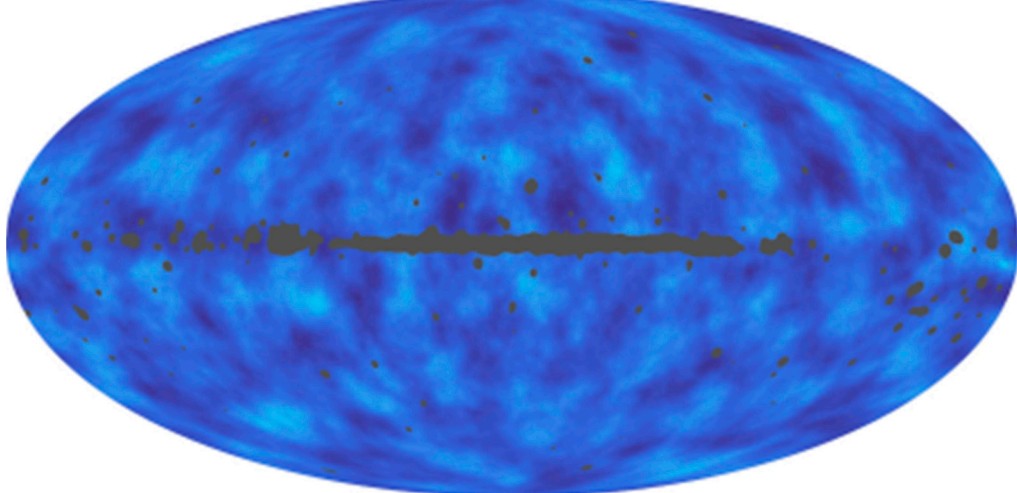

**Figure 14.** All-sky map of the CMB lensing potential constructed from the Planck 2015 data. Lighter regions correspond to integrated overdensities, darker to underdensities. The grey regions, where Galactic and extragalactic foregrounds are large, are masked in the analysis. Image credit: ESA and the Planck Collaboration.

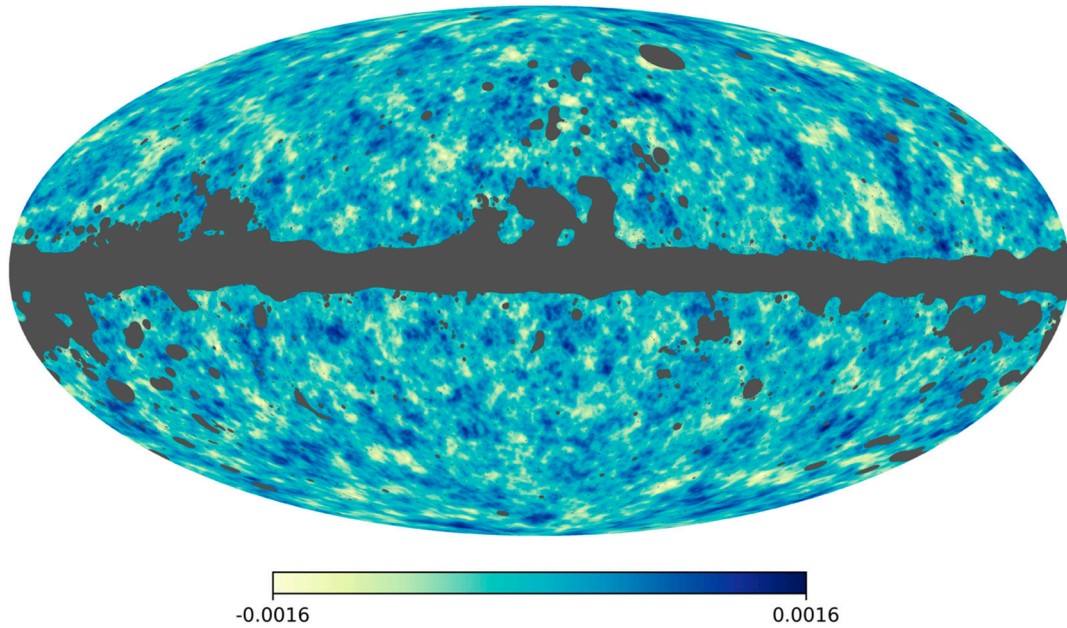

**Figure 15.** The 2018 Planck lensing deflection reconstruction derived from temperature and polarization maps. Dark blue areas represent regions that are denser than the surroundings, and bright areas represent less dense regions. The grey portions of the image correspond to patches of the sky where foreground emission, mainly from the Milky Way but also from nearby galaxies, is too bright, preventing cosmologists from fully exploiting the data in those areas. Image credit: ESA and the Planck Collaboration.

To close this section, we can't pass over the relevant observation of 1E 0657-56 (Bullet cluster) which is the result of the collision of two clusters of galaxies that happened there 150 million years ago and that may have lifted the veil between ordinary and dark matter, providing a very convincing evidence that the latter must exist. The study was reported in August 2006 by Douglas Clowe of the University of Arizona and a team of astronomers using the Chandra X-ray Observatory [109]

The baryonic mass of galaxy cluster is mainly gas and to a minor extent mass of stars that form separate galaxies. The mass of the latter makes up roughly 1 to 2 percent of the total cluster baryonic mass. These galaxies are immersed in a cold, low density gas and in a fully ionized hot plasma (10 to 100 million degrees K) visible at X-rays frequencies.

When clusters collided, stars passed through easily without incident due to the vast distances between them and therefore practically the star component of galaxies in the clusters was only gravitationally affected by the collision. Likewise, dark matter behaved as stars, not electromagnetically interacting, it crossed through the other dark matter smoothly. In contrast, the fluid-like X-ray emitting plasmas collided and lagged behind. The stellar component and dark matter continued on their trajectory and the fluid-like X-ray emitting plasma were spatially segregated. This collision caused the shock wave that can be seen in the bullet-shaped cloud of gas shown in Figure 16.

Using ground-based images of the Magellan and ESO telescopes and HST/ACS and Chandra X-ray surveillances of the cluster cores, Clowe and collaborators created gravitational lensing maps which showed that the gravitational potential does not trace the plasma distribution, which is the dominant baryonic mass component, but rather approximately traces the distribution of galaxies. This clear separation of dark matter and gas clouds is considered a clear and direct evidence of the existence of dark matter [109].

By the end of the eighties, gathered evidence of gravitational lens phenomena was by then sufficient to accept as normal to encounter lensed images. By then a number of other lensing phenomena have been discovered: For example, Einstein rings, quasar microlensing, galactic microlensing

events, and weak gravitational lensing. At present, literally hundreds of individual gravitational lens phenomena are known. Next, we shall describe some historical aspects of some of them.

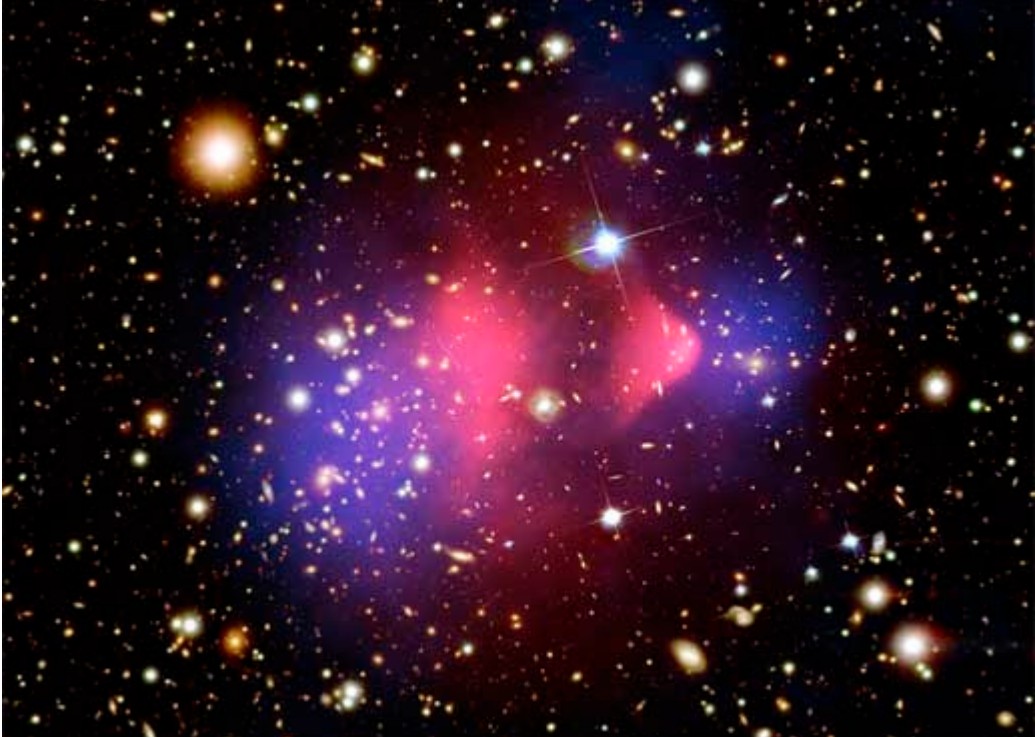

**Figure 16.** This composite image shows the galaxy cluster 1E 0657-56, also known as the "bullet cluster." This cluster was formed after the collision of two large clusters of galaxies. Hot gas detected by Chandra in X-rays is seen as two pink clumps in the image and contains most of the "normal," or baryonic, matter in the two clusters. The bullet-shaped clump on the right is the hot gas from one cluster, which was shocked in the collision but eventually passed through the hot gas from the other larger cluster during the collision. An optical image from Magellan and the Hubble Space Telescope shows the galaxies in orange and white. The blue areas in this image show where astronomers find most of the mass in the clusters via gravitational lensing. Most of the matter in the clusters (blue) is clearly separate from the normal matter (pink), giving direct evidence that nearly all of the matter in the clusters is dark. Image credit: X-ray: NASA/CXC/CfA/M.Markevitch et al.; Optical: NASA/STScI; Magellan/U.Arizona/D.Clowe et al.; Lensing Map: NASA/STScI; ESO WFI; Magellan/U.Arizona/D.Clowe et al.

## 16. Einstein-Chwolson Rings

As it was mentioned earlier in this saga, when the distant source, the lens galaxy and the telescope are exactly aligned, a ring image of the source is formed. This lensing effect was predicted by Orest D. Chwolson in 1924 and Einstein also pointed out this effect in 1936.

In 1988, two years after the first giant arc was reported in the vicinity of the galaxy cluster Abell 370 an extended radio source with the astronomical designation MG1131+0456 was discovered. This finding turned out to be the first example of a nearly complete "Einstein Ring", with a diameter of 1.75 s of arc. The discovery was made by a team of astronomers, using the Very Large Array [110]. However, the first complete Einstein ring was to be discovered not until 1998, over 74 years after its prediction [111]. This ring was found by a team of British astronomers using a 200 Km-wide MERLIN radio array consisting of a network of six radio telescopes spread out across England in combination with observations using the Hubble Space Telescope. The initial observations by MERLIN were followed up by a Hubble's detailed picture of the object and this revealed they made a spectacular "bulls-eye", i.e., they observed for the first time a complete Einstein Ring (Figure 17). The lens in this

case is an old elliptical galaxy, and the image we see is a dark dwarf satellite galaxy, which otherwise we could not see with current technology. Two further examples of almost perfect alignment are shown in Figures 18 and 19.

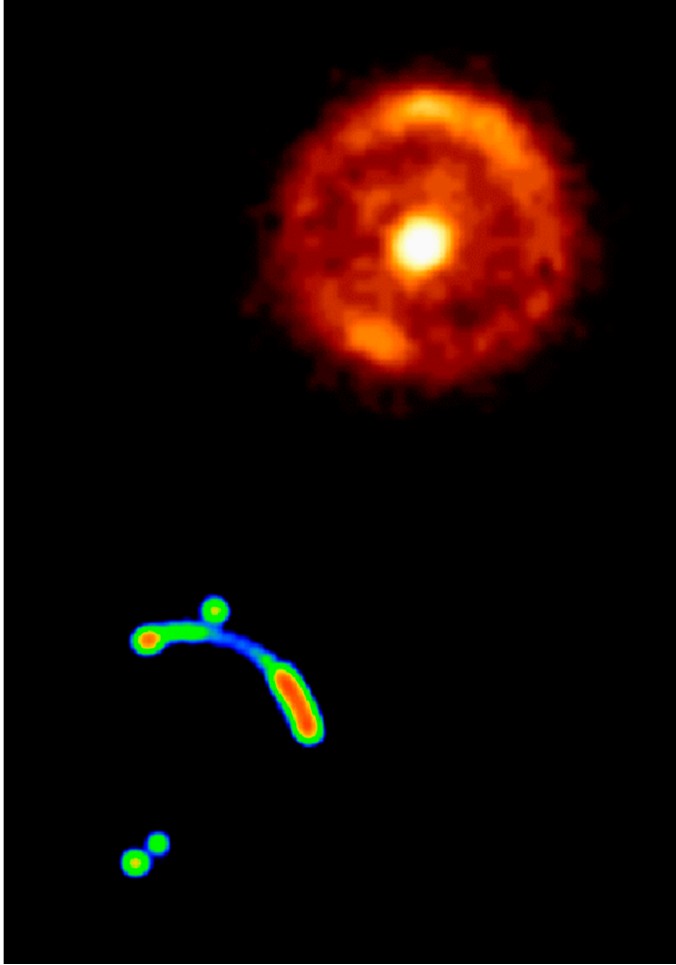

**Figure 17.** (**Upper**) The Hubble Space Telescope picture of the distant galaxy 1938+666 which has been imaged into an Einstein ring by an intervening galaxy. The intervening galaxy shows up as the bright spot in the center of the ring. The picture was taken in the infrared region of the spectrum and the computer-generated color of the image has been chosen simply for ease of viewing. (**Lower**) The MERLIN radio picture of the radio source 1938+666 embedded in the distant galaxy. The incomplete ring (or arc) shows that the radio source is not perfectly aligned with the lens galaxy and the Earth. The lens galaxy does not contain a radio source and hence does not show up in this picture. The colors are computer-generated and represent different levels of radio brightness. Credit: Neil Jackson, http://www.merlin.ac.uk/press/PR9801/press.html.

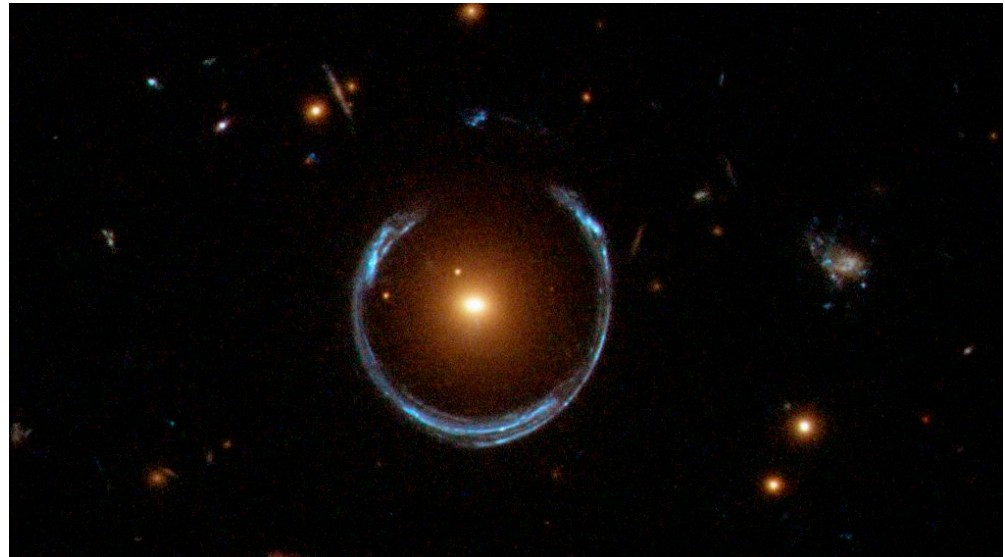

**Figure 18.** What is large and blue and can wrap itself around an entire galaxy? A gravitational lens mirage. Pictured above, the gravity of a luminous red galaxy (LRG) has gravitationally distorted the light from a much more distant blue galaxy. More typically, such light bending results in two discernible images of the distant galaxy, but here the lens alignment is so precise that the background galaxy is distorted into a horseshoe—a nearly complete ring. Since such a lensing effect was generally predicted in some detail by Albert Einstein over 80 years ago, rings like this are now known as Einstein Rings. Although LRG 3-757 was discovered in 2007 in data from the Sloan Digital Sky Survey (SDSS), the image shown above is a follow-up observation taken with the Hubble Space Telescope's Wide Field Camera 3. Strong gravitational lenses like LRG 3-757 are more than oddities—their multiple properties allow astronomers to determine the mass and dark matter content of the foreground galaxy lenses. Image Credit: ESA/Hubble & NASA (citation from APOD).

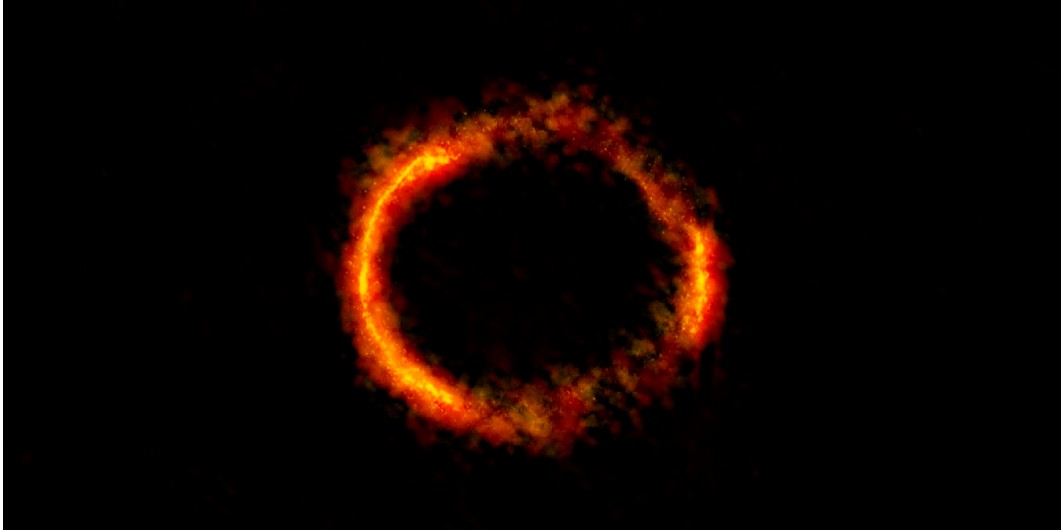

**Figure 19.** Gravitationally lensed galaxy SDP.81 taken by ALMA radiotelescope. The bright orange central region of the ring (ALMA's highest-resolution observation ever, the emission tracing a radius of ~1.5″) reveals the glowing dust in this distant galaxy. The surrounding lower-resolution portions of the ring trace the millimetre-wavelength light emitted by carbon dioxide and water molecules. **Credit:** ALMA (NRAO/ESO/NAOJ); B. Saxton NRAO/AUI/NSF.

## 17. The Hubble Parameter

In multiple imaged quasars the observer, the lens, and the source are not aligned so light rays that reach the observer from the source take different paths and therefore light rays arrive at different times. In addition, the gravitational field of the lens introduces a further time delay, the Shapiro delay [112], caused by spacetime dilation, which increases the path length in accordance to the field strength.

As early as 1964, Sjur Refsdal analyzed the gravitational lens equations and realized that it would be possible to determine the distance to a double-image quasar if one could establish the mentioned total time delay of the ray forming the second image with respect to the arrival of the first. But what concerns us here is that Refsdal also pointed out the possibility of measuring the Hubble parameter ($H_0$) by combining the value of the distance to the quasar obtained with his suggestion, with the quasar redshift distance [66].

After Walsh's discovery in 1979 that the supposed "double quasar" (0957+561AB) was actually a lensed system, attempts to measure Hubble's parameter using Refsdal's suggestion began in earnest. However, for many years these efforts were hindered by both the paucity of known lens systems and the difficulty in measuring time delays in the lensed systems.

The first individual to accurately measure the time delay between the pair of images was Rudolph "Rudy" Schild in 1986 [113]. His measurements were based on the careful observation of the double image of quasar 0957+561, the same quasar that five years before (in 1979) had been discovered by Walsh and colleagues [78]. By then it was known that the images of this quasar showed brightness variability, with brightness changes of a few percent occurring in a month.

Schild's procedure was to observe the pattern of brightness fluctuations in one of the images, and then cross-correlate this pattern with that of the second arriving image. In this way the obtained time shift between the two similar twinkling patterns is the sought time delay (see for example Figure 20).

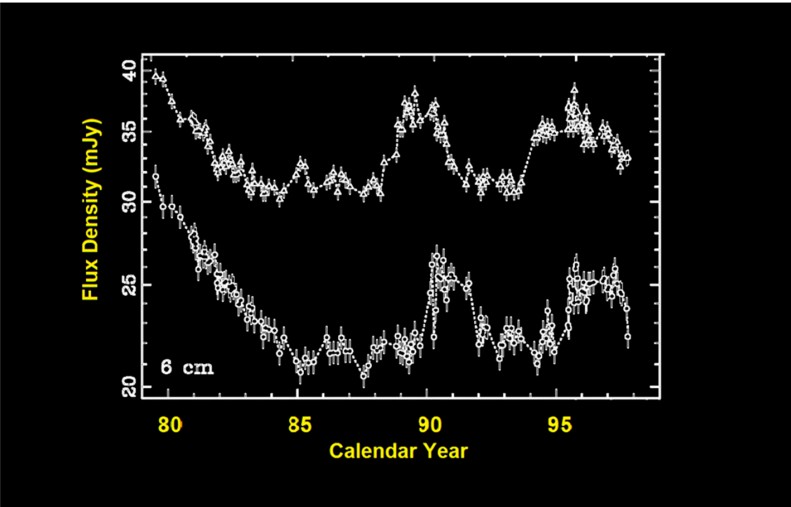

**Figure 20.** Monthly observations from 1979 to 1997 at 6 cm wavelength of the flux densities of the two images of quasar 0957+561 using the Very Large Array radio telescope (VLA) (After [114]). Fitting for the time delay between the images produced a value of 409 ± 30 days consistent with that obtained previously with optical wavelengths.

Rudy Schild and his student Bryan Cholfin recognized that the variation in the quasar discovered by Walsh, repeats itself in the second arriving image after 1.03 ± 0.01 years (376 ± 3.5 days) [113]. This value was confirmed by a French team in 1989 [115]. Using the time delay value measured by Schild and the mathematical models describing the lens positions and the gravitational fields, the estimated of $H_0$ at that time (1989) turned out to be 64 km s$^{-1}$ Mpc$^{-1}$. However, this result was far from what people expected at that time, so it was questioned and even ignored [116].

It is interesting to remark that the method suggested by Refsdal to obtain the Hubble parameter is simple and elegant, nonetheless the method depends on a good estimate of the shape and mass distribution of the lensing object to model its gravitational field which, as it was already mentioned, is responsible for bending and reducing the apparent light speed of the rays, and thus a realistic model is critical for an accurate estimate of the time delay. Another limitation of the Refsdal method is that most quasars brightness variations are no noticeable at human time scales, which makes it impossible to correlate the patterns of variability.

Nevertheless, what was not unnoticed was the obvious advantage that the time delay method to measure $H_0$ has over customary "distance ladder" methodologies to determine the value of $H_0$. In effect, the former method gives one step estimate of $H_0$ without the propagation of errors inherent in the distance ladder approaches. This motivation advantage set off many photometric monitoring campaigns to identify lensed quasars and measure their time delays.

These campaigns lasted several years since, on the one hand, lensed quasars are not abundant and on the other, brightness variations they present can be very slow and therefore cannot be perceived during prolonged observation periods or even a lifetime, as we have already pointed out.

One of the earliest campaigns was the COSmological MOnitoring of GRAvItational Lenses (COSMOGRAIL) collaboration [117]. In April 2004, COSMOGRAIL started a long-term photometry survey using five telescopes (1 m-class and 2 m-class) to obtain well sampled light curves of as many lensed quasars as possible suitable for a determination of $H_0$. By 2014 COSMOGRAIL had derived accurate time delays of around 20 lensed images of quasars [118].

In 2014 the H0LiCOW collaboration ($H_0$ Lenses in COSMO-GRAIL's Wellspring) benefits from the efforts of COSMOGRAIl 13-year light curves studies and the National Radio Astronomy Observatory Very Large Array monitoring and uses their data to estimate $H_0$ trying to reduce critical error sources, aiming to a precision of a few percent accuracy so to be competitive with other methods [119]. For a good estimate of the shape and mass distribution of the lensing object to model its gravitational field, HOLiCOW collaboration has been using throughout the years, a combination of ground- and space-based telescopes that includes the Hubble Space Telescope (HST), the Spitzer Space Telescope, the Subaru Telescope, the Canada-France-Hawaii Telescope, the Gemini Observatory, and the W. M. Keck Observatory. In particular, observations reveal Einstein rings in the lens systems that allowed HOLiCow researchers to perform precision lens mass [119].

Very recently, the 10th of July 2019 HOLiCOW presented a measurement of the Hubble constant from a joint analysis of six gravitationally lensed quasars with measured time delays; for a similar system see Figure 21 [120]. HOLiCOW found $H_0$ to be $[73.3]^{+1.7}_{-1.8}$ km s$^{-1}$ Mpc$^{-1}$, a 2.4% precision measurement. This result agrees with other previous measurements of this value (72.5 kms$^{-1}$ Mpc$^{-1}$ [121] and 74.2 kms$^{-1}$ Mpc$^{-1}$ [122]) from late time in the universe data but does not agree (the commonly used euphemism is "are in tension") with measurements of $H_0$ obtained using the PLANCK cosmic microwave background (CMB) radiation ($H_0 = 67.4 \pm 0.5$ km/s/Mpc) [123].

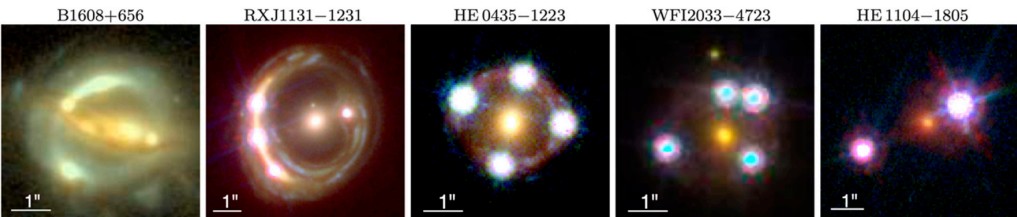

**Figure 21.** Multicolor images of the four quadruply lensed quasar systems in various configurations and one doubly lensed quasar system. The lens name is indicated above each panel from the H0liCOW collaboration; figure taken from [120]. Strong gravitational lens systems with time delays between the multiple images allow measurements of time-delay distances, which are primarily sensitive to the Hubble constant. They calculated that the Universe is actually expanding faster than expected on the basis of our cosmological model.

## 18. Other Recent Developments

As we mentioned, lensing is now a very active field of research, so it results inconvenient to historically review new-born topics that are currently being developed. There are few that we would like to briefly mention, though, just to show the richness of this exploding science area.

### 18.1. CMB Lensing

Early universe photons interact with protons and electrons until the universe becomes recombined, when protons and electrons match together to form atoms. That happened when the universe was approximately 380,000 years old, at the last scattering surface. Afterwards, photons do not experience many interactions and travel freely in an expanding universe. These photons are referred as Cosmic Microwave Background (CMB), since their intensity is highest at that wavelength when we measure it. Lensing happens due to the gravitational potential wells that CMB photons encounter in its way from the last scattering surface to us (here and now).

Where do potential wells come from? After matter-radiation equality dark matter begin to accrete into structures, triggered by inhomogeneities originated thanks, so the standard lore, to quantum field perturbations during inflation. Dark matter disturbances then develop halos, and baryons after the universe recombines at $z \sim 1100$ fall into these gravitational potential wells to form galaxies and clusters that are in-between the last scattering surface and us. Thus, photons traveling towards us, feel these potentials and get gravitationally lensed inducing anisotropy correlations of the CMB maps, see Figure 22.

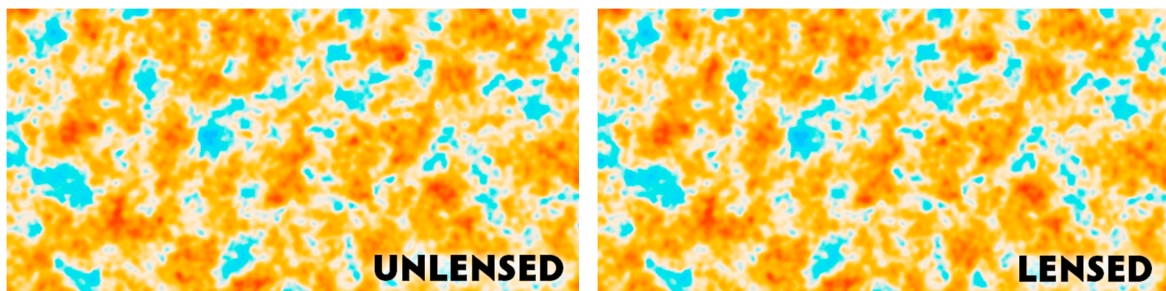

**Figure 22.** CMB Lensing: This illustration shows how CMB photons are deflected by the gravitational lensing effect of massive cosmic structures as they travel across the Universe. Image credit: ESA/NASA/JPL-Caltech. To appreciate in more detail the differences see a blinking gif at https://www.nasa.gov/mission_pages/planck/multimedia/pia16880.html.

In fact, potential wells yield three effects on CMB photons: a change in temperature pattern of hot and cold spots around foreground masses, generation of non-Gaussianities, and a twist of the polarization plane of CMB photons, bringing original E-modes into the so-called lensed B-modes (not to be confused with B-modes caused by gravitational waves, see Figure 23). The effect of lensing in CMB photons is to broaden and attenuate the acoustic peaks of its power spectrum (see Figure 24) at angular scales less than 2.4 arcmin and these effects can be used to constrain cosmological parameters given CMB data[2]. Historically, the first theoretical treatment on the subject was made by Eric Linder [124], in which he discussed that CMB is affected by density fluctuations causing two effects: large scale shrinking of angular scales and small scales smearing. The first direct measurement of the power

---

[2] A typical linear potential for the large-scale structure is of the order of $\psi \sim 2 \times 10^{-5}$ implying a deflection of $\beta \sim 4\psi \sim 10^{-4}$ radians. In the overall path of nearly 14,000 Mpc, there is a structure of this scale about every 300 Mpc giving approximately 50 deflections. This trajectory is a random walk and the RMS deflection is roughly $50^{1/2}\beta \sim 2.4$ arcmin. The CMB lensing can be fully described via the deflection field: $\Theta(\hat{n}) = \hat{\Theta}(\hat{n} + \nabla\psi)$, where $\Theta$ and $\hat{\Theta}$ are the lensed and unlensed angles, respectively.

spectrum of the CMB lensing potential was reported in 2011 by the Atacama Cosmology Telescope collaboration [125]. Also, this information was further used to provide another evidence for dark energy, but this time from the CMB alone. Later it was also reported a similar measurement by the South Pole telescope data [126]. However, it is fair to say that prior to these measurements there were indirect detections of CMB lensing cross-correlated with tracers of the lensing masses using CMB WMAP data and radio galaxy counts from the NRAO VLA sky survey and using luminous red galaxies and quasars from the Sloan Digital Sky Survey [127,128]. Later on, in 2013 the South Pole Telescope (SPTpol) collaboration reported the first detection of gravitational lensing B modes [129], and in 2014, ACTPol reported [130] a reconstructed CMB lensing potential using temperature and polarization data in cross-correlation with the cosmic infrared background. Figures 13–15 show dark matter reconstructions through gravitational lensing of all sky CMB maps. The Planck collaboration was the first to revel the lensing maps of the entire sky for first time. Previous experiments covered on small patches of the sky.

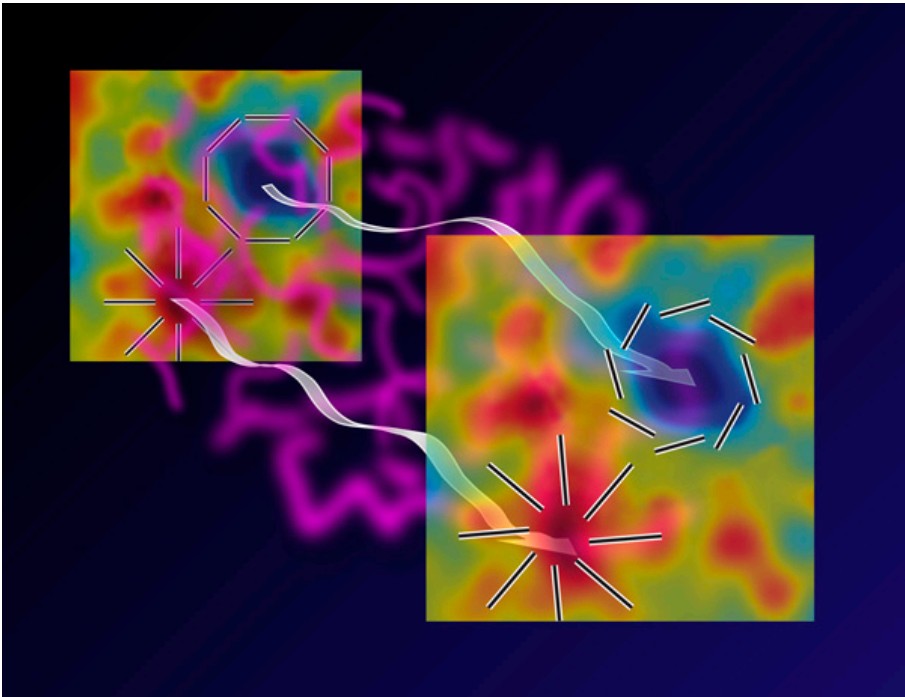

**Figure 23.** Twisting E-mode to B-mode: When the cosmic microwave background (CMB) originated 13.7 billion years ago (represented by the map on the left), it was polarized with radial and tangential patterns around hot (red) and cold (blue) regions, respectively. This E-mode polarization has been previously observed in CMB measurements. However, gravitational lensing from intervening matter (purple) causes a slight twist in the primeval pattern (shown in the map on the right). This B-mode polarization has been detected for the first time by the SPT Collaboration. (Credit APS/Alan Stonebraker, https://physics.aps.org/articles/v6/107).

We see then that CMB lensing is now a standard method to extract cosmological information, as was also shown in the PLANCK collaboration paper in 2018 [131]. To conclude on this theme, we refer the reader to a list of historic and recent CMB experiments that is given in [132].

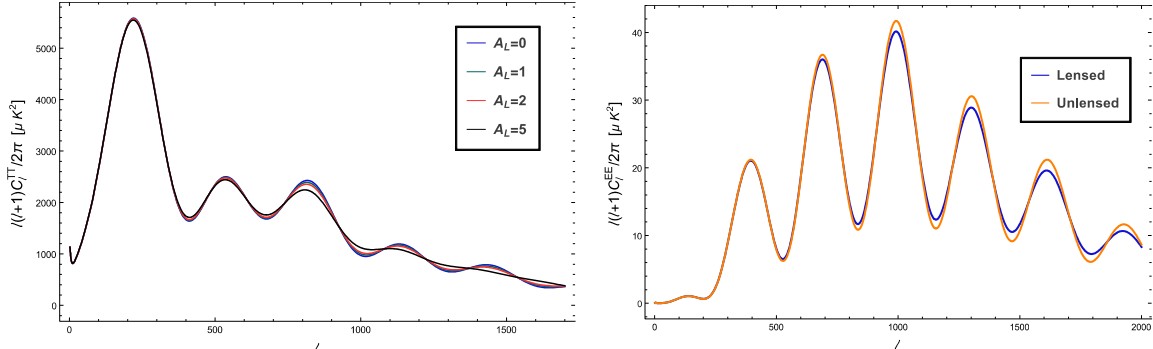

**Figure 24.** Gravitational Lensing Smears CMB power spectrum peaks both temperature (left panel, for different lensing amplitudes) and E-mode polarization (right panel): Again gravitational lensing from intervening matter (purple) causes varying across the map slight magnification variation in the primeval pattern effectively averaging over several maps and thus smoothing down the peaks. Magnification is equivalent to stretching the power spectrum. (Credit Alejandro Aviles).

## 18.2. Microlensing

Another lensing possibility is microlensing, that happens when a distant point source is lensed by a point mass, creating two images of the same source, whose brightness is more intense than the original source's. If the resolution is not enough to resolve the both images, one sees the event as a change in the apparent brightness of the source. This method has been discussed by many authors, starting as above mentioned with Einstein in 1936, later independently by Liebes and Refsdal in 1964, and others later on. Microlensing has been proposed to detect dark objects in a broad mass range, from Earth-sized planets to black holes and neutrons stars. One example of using this technique was applied to MAssive Compact Halo Objects (MACHOs) in the galactic halo by Paczynski in 1986 [133], and it was discussed in more detail by others in the 1990s. It was thought that MACHOs could account for the dark matter in halos, but by now it is clear that they cannot be a significant fraction of the entire mass of halos. The microlensing technique has been also suggested by Paczynski [134] and Griest in 1991 [135] to detect brown dwarfs and other objects whose mass is much less than our Sun's mass in the Galactic disk and bulge, among other applications.

Another example of recent use of microlensing is to search extra-solar planets, exoplanets. It happens when the star that host the exoplanet acts as a lens, but the presence of the latter induces an extra burst of brightening in the resulting lensed light curve. In this way, one can measure the mass of the exoplanet and the distance to its companion star. The technique was originally proposed by Shude Mao and Bohdan Paczynski in 1991 [136] to see effects on binaries, a star-star system, or even a star-planet, and was refined by Andy Gould and Abraham Loeb in 1992 to detect exoplanets [137]; later the Optical Gravitational Lensing Experiment (OGLE) implemented the technique in real observations making it feasible. The method can detect far-away planets, e.g., towards the center of the galaxy, where it results convenient since the galactic bulge provides a large number of stars. Microlensing is especially good at detecting low mass planets in wide orbits or free-floating (that have been ejected from their parent systems), where other methods are ineffective, and it has a high signal to noise ratio. These properties make microlensing the ideal method to detect Earth-like planets around Sun-like stars. One of the drawbacks is that microlensing captures unique events, so it might be hard to repeat measurements, so its confirmation may be employing other techniques. There are nowadays some observatories using this technique, and moreover there is a network of telescopes to follow up microlensing events providing accurate light curves and indicating whether a planet is present or not. There are, as of today, already more than four thousand exoplanets discovered using this and other techniques, the most by using transit photometry.

Another instance where microlensing appears is when a very distant source is lensed by a galaxy and as the lensed object transits near a caustic line, then individual massive stars can distort the caustic

or effectively microlens modulate an image that is already highly magnified on a relatively short time scale. This has been used to search for primordial black holes at the 30 $M_\odot$ level as well as image very distant galaxies and quasars. Also, this microlensing technique has been successfully used to study inner structures of AGNs, from the early evidence in 1989 [138] to the present [139–143]. It seems promising that the method will provide a significant boost of our understanding of the AGNs structure, as well as black hole mass and spin thanks to the predicted increase of the number of such systems known. Today, we know a few dozens of bright four-image microlensed quasars systems, but by the mid-2020s, due to wide-field surveys including DES, DESI, LSST, and Euclid, hundreds even up to thousands of suitable systems are expected to be found [144].

### 18.3. Black Hole Shadow

The basics of black hole (BH) astrophysics was developed in the early 1970s. Understandably and without delay, physicists speculated on how to detect them if they really existed. In 1972 P. J. E. Peebles gave a general account on how, at that time, black holes might be detected through their long-range Newtonian gravitational field in globular clusters, galactic nuclei and in other sites [145]. But others realized that detection attempts would have to be based on observations of BH's gravitational effects on matter in their near vicinity or radiation lensing. The next following year (1973) James M. Bardeen and his then student C.T. Cunningham published a paper entitled, "The optical appearance of a star orbiting an extreme Kerr Black Hole" [146]. In that paper they analyzed both: the energy flux emitted by a star in a circular orbit in the equatorial plane of a BH and its apparent sky position as it might be seen by a certain distant observer. Almost simultaneously N. I. Shakura and R. A. Sunyaev, addressed the same question on the observational appearance of BH [147]. These authors had already accepted that most BH were possibly surrounded by gaseous material forming a hot accretion disk which emits a characteristic spectrum of electromagnetic radiation [148]. Shakura and Sunyaev in their exploratory study just mentioned, considered the effects of angular momentum transfer between both bodies (accreting ring and BH) on the radiation that the BH-disk system emits under different conditions. A year later, in 1974 Bardeen expanded his previous research commenting on some properties of BH relevant to their observation [149].

But it was only until 1978, that following a suggestion from his former PhD advisor Brandon Carter, the young French researcher Jean-Pierre Luminet wondered what a Schwarzschild BH surrounded by a luminous accretion disk would look like. Luminet got to work on the problem and the following year he published his results [150]. He presented the appearance of the accretion disk gravitationally lensed by a non-spinning Schwarzschild BH, as seen from far away but near enough to resolve the image.

By calculating photon trajectories in the BH gravitational field within a geometrical optics approximation, he constructed a simulated photograph of the BH-accretion disk arrangement as seen by a distant observer at 10° above the disk's plane. To get the simulated image, Luminet used an up-to-date (1970's) transistor-built computer with punch-card inputs to obtain data of several thousand photon trajectories. Then he patiently drew dots by hand on a sheet of white paper and thus formed an image of the lensed accretion disk. Next, he manually took a photo of the result and thus formed the image shown in Figure 25. It is clear that the image shown in the figure displays a strong asymmetrical photon flux distribution.

This asymmetry is due to two effects. The first effect results from the very strong gravitational lensing that enormously bends the trajectories of photons emitted from the accretion disk sector located behind the BH. These photons are so strongly lensed that their image appears above the BH. This might seem weird. It would be like seeing the hidden back sector of Saturn's rings, when viewed at a small angle above their plane, on its top. The second effect is a Doppler shift produced by the accretion disk rotation around the BH. Those photons emitted from the part of the disk moving away from the observer are red-shifted and in those from the approaching part, blue-shifted.

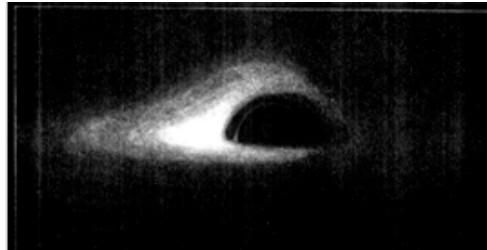 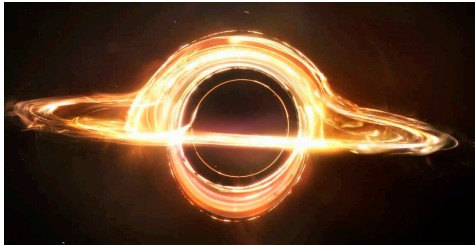

**Figure 25.** On the left, simulated photograph of a spherical black hole with a thin accretion disk, as seen by an observer 10° above the disk's plane. Picture taken from [150]. On the right, in 2014, Christopher Nolan released the movie Interstellar and his science advisor Kip Thorne and colleagues used theoretical equations in order to approximate what a black hole could look like near maximal spin [151].

During the 1970s evidence began to emerge about the real existence of BHs. The discovery of a very intense and compact Sagittarius A* radio source (Sgr A*) at the center of our own galaxy seemed to be the definite indication that a very massive BH would be located there [152]. In addition, it had been suggested that the giant elliptical galaxy M87 could also harbor a giant BH with of few billion solar masses ($\sim 5 \times 10^9$ M$_\odot$) [153].

Curiously enough, eight years after the discovery of the BH in the Sagittarius sector of the sky, one of its discoverers (Robert L. Brown) coined the name Sgr A* for this BH. This happened while he was thinking of the radio-source he co-discovered as "*the exciting source*" of the nearby hydrogen clouds making them glow. In his own words: "*When I began thinking of the radio source as the exciting source. . . the name Sgr A* occurred to me by analogy brought to mind by my PhD dissertation, which is in atomic physics and where the nomenclature for excited state atoms is He*, or Fe* etc.*" [154].

The pair of BH observations just mentioned, looked as if they would immediately trigger a campaign to observe these BHs, but the adequate technology for the observation of active galaxies nuclei (AGN), with the necessary resolution, was still lacking in those times. Meanwhile, during the following years, BH image simulation raised almost null interest, and that only as a mere educational exercise or academic curiosity [155,156]. For a very detailed historical description of the simulations that followed, we refer the reader to the Luminet's review papers [157,158]. A modern view of the Kerr black hole photonsphere is given in e.g., ref. [159].

On the other hand, from measurements of gas and star dynamics, evidence had been accumulating over the years that a central dark mass should reside in the center of our galaxy. As a matter of fact, A. Eckart and R. Genzel reported in 1997 a surveillance made on motions of several stars in the innermost core of our Galaxy [160]. From the dynamics of the observed stars they concluded that there is a central dark mass located in Sgr A*. Their mass density data excluded the fact that the central mass concentration was in form of a compact white dwarf or neutron star cluster. Consequently, they inferred that central mass concentration is a single massive BH in SGrA* with a mass of approximately four million solar masses.

The turning point of our narrative came in the year 2000 when the possibility of viewing in practice the shadow of Sgr A* with an array of radio telescopes was first discussed in a paper by Heino Falcke, Fulvio Melia and Eric Agol [161]. These authors pointed out that the angular diameter of the BH shadow of Sgr A* as seen from Earth would be around 30 microarcseconds (μas) considering gravitational lensing of light due to the BH. They made this calculation with a General-Relativistic ray-tracing program. The program predicted a shadow identical in shape but ten times larger than the event horizon. They also considered in their paper the suitable frequencies to observe the BH shadow taking into account interstellar scattering, atmospheric transmission and the telescope resolution for global Very-Long-Baseline Interferometry (VLBI) arrays of radio telescopes. They found that an appropriate observation wavelength to be of the order of sub-millimeters. They concluded that perhaps

with the next generation of VLBI at sub millimeter wavelengths it will be possible to image the shadow of the BH.

On the occasion of the 30th anniversary of the discovery of Sgr A*, a meeting was organized in Green Bank West Virginia. In addition to the regular program of talks, an informal discussion on future millimeter/submillimeter VLBI observations of Sgr A* was summoned. The intention of this gathering was to discuss the new opportunities that the prompt arrival of next generation of radio-telescopes would bring to imaging on the scale of the Sgr A* event horizon. The success of this unique observations would provide detailed test of accretion models and General Relativity [162].

During this informal gathering the three speakers: Sheperd S Doelleman, Heino Falcke and Geoffrey Gower stressed the scientific importance of imaging a BH shadow. They weighed what would be the technical requirements necessary to reach this goal. Some technical requirements and scientific goals were also discussed and finally it was realized that success will require joining efforts and considerable coordination of resources at a number of institutions and observatories [163].

By 2009 a Science White Paper was submitted by Doelleman to the ASTRO2010 Decadal Review Panels, launching the initiative to combine existing and planned mm/submm facilities into a high sensitivity, high angular resolution "Event Horizon Telescope", capable of imaging a black hole [164].

In the summer of 2017 a formal memorandum of understanding was signed by 13 stakeholder institutions with about 150 individual scientists to form the Event Horizon Telescope Consortium (EHTC), which currently is conducting observations with 8 telescopes (IRAM 30m, JCMT, SMA, SMTO, SPT, LMT, ALMA, APEX) [163].

The 10th of April 2019, the Event Horizon Telescope Consortium (EHTC) announced in a series of six papers published in a special issue of The Astrophysical Journal Letters the first image of a black hole at the center of the galaxy M87. Figure 26 shows the image. The discovery was made during the centennial year of the historic eclipse observation that first confirmed the theory Einstein's general relativity.

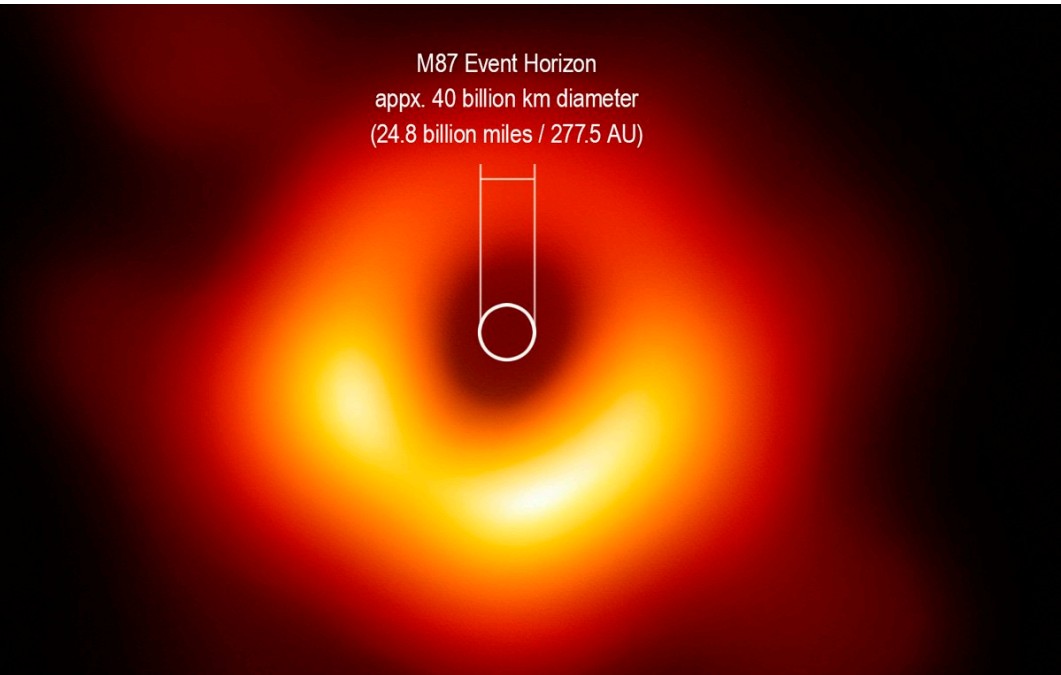

**Figure 26.** The shadow of the Black Hole at the center of the Galaxy M87. As shown, the even horizon is smaller than the resolution of the picture, so the detailed light paths in Figure 25 cannot be appreciated with the actual resolution of the Event Horizon Telescope. Figure taken from [165] and modified by https://twitter.com/JPMajor.

### 18.4. Gravitational Lensing of Gravitational Waves

Up to this point we have given in previous sections, an account of several different historical events on lensing of electromagnetic waves (EWs). Now we shall narrate the case of gravitational lensing of gravitational waves (GWs). But before we go into the lensing issue, it is convenient to recall some past episodes of GW research itself.

At the beginning of the second half of the twentieth century there were still doubts on the existence of GWs. Up until the Chapel Hill conference in 1957 there was significant debate about the physical reality of gravitational waves [166], since no experimental evidence existed by then. In the 1960's decade some scientists undertook GWs research to the experimental field. In this regard it is worth remembering Joe Weber's experiments on resonant bars built to detect GWs [167]. By the 1970s observations made with Weber bars were gradually discredited and the level interest in the experimental aspects of GW research declined [168]. This disenchantment is explicable since at that moment the possibility of detecting GWs seemed to be far away in time. In contrast, GWs theoretical research has always been latent since the end of the Second World War. Universities gradually began incorporating General Relativity in their curricula. By the 1960's gravitation lectures were regularly offered in major universities such as those given by Richard Feynman at Caltech in the 1962–1963 academic year. One of the topics treated by Feynman in those lectures was precisely GWs. In his lectures, he noted the similarities between gravitational and electromagnetic waves [169].

Other lectures followed, as the series given by Kip Thorne in 1987 in the occasion of the 300th anniversary celebration of the publication of Newton's "Principia". Lectures were collected in a book edited by Steven Hawking and Werner Israel [170]. In the chapter authored by Thorne, entitled "Gravitational radiation" one can read, "*Lumps of background curvature associated with black holes, stars, star clusters, and galaxies will focus gravitational waves in precisely the same manner as they focus electromagnetic waves; . . .* " [171]. In other words, gravitational lensing of GWs should occur in the similar way as it does for light. However, one must bear in mind that there are different approaches to GWs focusing analysis, basically the geometrical approach and the wave approach. They depend on values of the lens mass and GW radiation frequency.

Now, two brief remembrances before we go directly to the topic of GWs lensing: In 1974 the discovery of a binary pulsar by Taylor and Hulse placed a renewed attention in GWs as this event gave certainty of GWs reality [172]. And in the 1990s interest in GWs was further boosted when it was glimpsed the strong possibility that LIGO-type projects that were developed at that time, had a successful end. For a historical review on GWs see Cervantes-Cota et al. [173].

It was then in the 1990's when studies directly related to gravitational lensing of GWs started to appear in the literature. Possibly the first work (1996) related to LIGO-type facilities was authored by Yun Wang, Albert Stebbins and Edwin Turner [174]. Their intention was to estimate the number of merging neutron star binaries that might be observed within a year of operation. Their analysis took into consideration specifications and observational constrains of proposed advanced LIGO-type gravitational wave detectors. In their analysis they considered two types of lensing: macrolensing due to galaxies and microlensing due to compact objects such as stars. Taking into account the redshift distribution of both groups (galaxies and compact objects) and making an assumption on the fraction of matter belonging to each of the two groups, they estimated the expected total number of events. Their calculation yielded that an advanced LIGO should see more than five strongly lensed events per year if the matter fraction in compact lenses was close to 10%. They maintained that this significant number of inspiral events can be detected because gravitational lensing magnifies GWs.

In 1999 another paper of gravity wave lensing appeared in the literature. This time the author Anthony A. Ruffa considered the center of the Milky Way acting as a gravitational lens that focuses gravitational wave energy to the Earth from a distant GW source located very near the line that passes through both, the galaxy center and the Earth's center [175]. As example of application, Ruffa chose as the source of GW, a rotating neutron star outside our Galaxy emitting a GW of 60 Hz. To solve the problem Ruffa employed the wave approximation and concluded that the amplification of GWs due

to lensing is huge. It is worth mentioning that the author chose to position: the source, the galaxy's center and the Earth's center on the same line, a highly unlikely situation.

Few years later in 2002, an article criticizing the 1987 pioneer paper by Wang et al., appeared in the literature. The paper was authored by A. Zakharov and Y. Baryshev, [176]. There they indicate that Wang et al. used the geometrical optics approximation model for gravitational lensing which is adequate for lensing by extended objects such as galaxies, but not for gravitational lensing of stellar objects of stellar masses. Zakharov and Baryshev showed that the application of the geometric model leads to an over estimation of the observable events rate. Almost simultaneously, in 2003 a paper by R. Takahashi and T. Nakamura arrived at the same conclusions that Zakharov and Baryshev, had reached before, about Wang's et al. geometrical approach [177]. They stressed that in the gravitational lensing of GWs, the wave optics should be used instead of the geometrical optics when the wavelength λ of the GW is greater than the Schwarzschild radius of the lens mass.

Most of the articles that were published later, dealt with estimates of detectable events rates that would probably be seen in the sky over a year, when LIGO-type facilities were fully operating. Knowledge of this subject matter was understandable has its knowledge may provide clues on the evolution of the early universe. Other articles addressed different lensing topics such as the one published by G. Bimonte et al. on cosmological waveguides for GWs [178]. There they explore the possibility that a gravitational wave from a distant source gets trapped by the gravitational field of a long filament of galaxies of the kind seen in the large-scale structure of the Universe. These authors hypothesized that such a waveguide effect could lead to a huge magnification of the radiation flux from distant sources.

On 14 September 2015 a GW signal was detected of two ~30 $M_\odot$ BHs merging about 1.3 billion light-years from Earth by LIGO laser interferometers in Hanford, Washington, and Livingston, Louisiana of the LIGO Scientific Collaboration [179]. After this first detection LIGO together with VIRGO, have together so far announced observations of several further GW events from ten BH-BH mergers as well as one binary neutron star inspiral. Most of these events involve merging of bulky BHs having total masses from 18.6 $M_\odot$ to 85.1 $M_\odot$. In addition, BH-BH mergers occurred in the relatively nearby universe, in a range in distance between 320 Mpc and 2750 Mpc from Earth [180]. As it is known, the importance of measuring gravity waves deserved the Physics Nobel Prize in 2017 to Rainer Weiss, Barry C. Barish and Kip S. Thorne. This and the discovery of binary neutron star merger GW170817 [181] by LIGO and its associated electromagnetic counterparts marked a new-borne area of multi-messenger gravitational wave astronomy that is increasingly active. As an example, the almost simultaneous signal arrival of GWs and EWs from the neutron star binary GW170817, with a slight delay of less than 2 s, helped to constrain alternative, modified theories of gravity that have different speeds or couplings of light and gravitational waves [182].

As the subject became trendy specific computations for detectors were demanded. For instance, Ryuichi Takahashi reported that the transition to wave optics occurs, if the lens mass is less than approximately $10^5$ $M_\odot(f/Hz)^{-1}$, where f is the GW frequency. The GW is predicted to be early by typically ~1 ms $(f/100\ Hz)^{-1}$ for ground-based detectors, ~2 min $(f/mHz)^{-1}$ for space-based interferometers, and ~4 months $(f/10-8\ Hz)^{-1}$ for pulsar timing arrays [183]; similar estimates are given in [184].

On the other hand, X-ray studies of black hole pairs in the Milky Way suggest their mass distribution hits its highest point around 10 solar masses [185]. If it is accepted that the same distribution applies to the BH population in the whole region of space that LIGO surveys, then the range of masses seen by LIGO and Virgo implies a local paucity of black holes in our near neighborhood or LIGO and Virgo are actually observing smaller merger events taking place much farther away, magnified and made visible through gravitational lensing [186].

One of the effects of gravitational lensing is to produce multiple images, as in the case of lensed quasars, and for ground-based detectors, such as LIGO/Virgo, there exists a degeneracy between a lensed GW from a high-redshift, low-mass source and an unlensed GW from a low-redshift, high-mass source. In addition, the images of lensed quasars do not arrive to us simultaneously but exhibit a

time delay (c.f. Section 17 the Hubble parameter, Figure 20). A similar effect may be happening with the BH-BH mergers signals. In simple terms, if two similar signals of BH-BH mergers are observed one after the other and in the same region of the sky, this could be a strong indication that we are detecting a lensed event. If the gravitational lens is a large elliptical galaxy the time delay can be on order hours/days to years for high magnification. That could be the case of a pair of BH-BH merger signals GW170809 and GW170814 that were registered as independent events, separated by a few days. Tom Broadhurst, Jose M. Diego and George F. Smoot III [187] put forward the hypothesis that these events could be originated by the same event since both possess indistinguishable waveforms and similar strain amplitudes, implying a modest relative magnification ratio, as expected for a pair of lensed gravitational waves. The LIGO collaboration is however skeptical of this interpretation. Time will tell when better data analysis is performed.

On the other hand, extra features as time delays and consequently interference distortions in signal can be introduced by microlensing [188] and by wave optics effects if GW are observed with sufficiently high signal-to-noise ratios.

Massive particles can also be gravitationally lensed though differently from EWs and GWs because their propagation speed is not the same as light but they are only a little slower for low masses. Thus, neutrinos are a prime candidate for this effect. In principle, a massive and massless particle will be deflected by slightly different amounts at the lens. The masses of left-handed neutrinos are so low ($m_\nu < 0.1$ eV) that even over cosmological distances ($z\sim5$), the time delay is less than a second. If the signal were strong and clear, it might be possible to see the neutrinos arrive in narrowly spaced delays corresponding to mass [189].

## 19. The Remains of the Eclipse

On 6 November 1919 at a meeting held in Burlington House, London, the Royal Astronomer sentenced: "*The results of the expeditions to Sobral and Principe can leave little doubt that a deflection of light takes place in the neighborhood of the Sun and that it is of the amount demanded by Einstein's generalized [sic] theory of relativity*".

From that moment the effects of gravitational lenses captured worldwide attention. Studies on these effects began to be developed. Here, in the present review, we have shown that first it happened at a slow pace and then it went through faster development until it reached an inflationary period. At that point we had to abandon the chronological narrative we were following, to adopt a thematic description. We recognize that it was not possible to address all possible topics. We left some out, just as we omitted some important names and works. These omissions were involuntary and we offer an ample apology. We chose to write this historical review devoid of formulae, as there are some excellent technical reviews and books in the literature, e.g., [190–193].

Future surveys will provide observations for a large number of gravitationally lensed sources, that will allow us to apply lensing to study a variety of astrophysical problems. The increase of a few orders of magnitude in the number of lensed systems will turn the next decade into an exciting era of gravitational lensing. A significant extension on the uses of strong lensing will be its practice as a method to infer precise locations of high energy emissions sources [194]. As we have already mentioned, gravitational lenses induce time delays and produce multiple images of sources. Both effects on lensed images can be used to drastically improve the angular resolution of current telescopes. In other words, gravitational lenses can in fact act as high-resolution telescopes. This will bring enormous consequences to astrophysics as will allow us to explore e.g., the inner regions of active galaxies. These regions host the most extreme and energetic phenomena in the universe including, relativistic jets, supermassive black hole binaries, and recoiling supermassive black holes. For an outlook on the future of this topic we refer the reader to a recent review [195].

In this sense, the most far-reaching scientific legacy of the 1919 Eclipse (Einstein's Eclipse as sometimes is called) is the ongoing vast development of Gravitational Lensing as a scientific tool, to heights no one had previously envisaged. This year we are celebrating the 100th anniversary of

"Einstein's Eclipse" an event that for the first time verified key paradigms of modern physics. Einstein, after being informed of the positive results of the eclipse, decided to celebrate by buying a violin. Everyone is free to choose how they celebrate. We did it our way, by writing this review.

**Funding:** J.L.C.-C. was funded by CONACYT grant number 283151.

**Acknowledgments:** This research has made use of NASA's Astrophysics Data System Bibliographic Services.

**Conflicts of Interest:** The authors declare no conflict of interest.

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
