# Peer review of "The Legacy of Einstein’s Eclipse, Gravitational Lensing"

_universe, doi:10.3390/universe6010009_

Round 1

Reviewer 1 Report

This is an excellent review on the topic of gravitational lensing. Starting from a historical perspective, describing the pre-history of the subject (before General Relativity) to the modern applications of lensing and their impact on cosmology and astrophysics, it provides an interesting reading with many useful references. I recommend it for publication without hesitation.

Author Response

Thank you for your comments.

Reviewer 2 Report

(see attachment)

Author Response

Dear Universe Editors, We have made the amendments suggested by reviewer 2 and reviewer 4 (the other two reviewers had no criticisms). These are as follows:    Reviewer 2:   Related to the first criticism: On lines 359-360 of the new version the phase “spacetime curvature in the presence of matter fields [26].” was changed to “both time and space contributions were now considered, whereas the first calculation took into account only the temporal curvature contribution [28]."   Related to the second criticism: On line 607 of the new version the phrase “...has no unique focal...” was changed to “...has not necessarily a unique focal…”, and in lines 613-617 we added “Though there are these differences, in practice one finds configurations that possess a well-defined focal plane, e.g. when one approximates a lens distribution as a sheet of uniform surface density, and in this case, the analogy to a lens is well approximated, except only for the spatial dependence of the lens refractive index. “   We corrected some typos and grammatical mistakes. We highlighted these changes in yellow, as well as some reference additions suggested by the editors. 

Reviewer 3 Report

This paper is a review of the story of the deflection of light, from the speculations of 18th century scientists to the present-day gravitational lens industry.  I enjoyed the review, and found it to be balanced and accurate, at least in the areas that I know something about.  I recommend publication in Universe.

Author Response

Thank you for your comments.

Reviewer 4 Report

“The Legacy of Einstein’s Eclipse, Gravitational Lensing” provides an in-depth historical overview of gravitational bending of light starting as a curiosity of a few individuals, through a proof of general relativity, to well-established tool in astrophysics used to elucidate the universe. Below are more detailed comments. 

I would like to highlight authors outstanding depth of historical knowledge, giving a credit to scientists that had a critical role in first observations of gravitational lensing, as well as, very accurate explanations that are often being omitted in the literature, like for example difference in the focal point for gravitational lensing and geometric optics approximation.

Line 46: “strong magnification of faint sources” -> sources don’t have to be faint to be magnified. Gravitational lensing magnifies all sources, which is used to detect faint sources that wouldn’t be observable otherwise. 

End of the paragraph at line 56. Authors list important applications of gravitational lensing. I would urge to mention using lensing to resolve structures of active galaxies from event horizon at radio, through accretion disc at optical and X-ray (https://iopscience.iop.org/article/10.1088/0004-637X/729/1/34/meta) to relativistic jets at gamma-rays (https://iopscience.iop.org/article/10.3847/0004-637X/821/1/58https://iopscience.iop.org/article/10.1088/0004-637X/788/2/139). It is worth emphasizing that gravitational lensing has been applied to the entire electromagnetic spectrum from radio to gamma rays, and allows us to study structure as small as Event Horizon and as large as kiloparsec relativistic jets. 

Line 74: Measuring redshift of quasi-stellar object 3C 273 and concluding it’s cosmic origin in 1964 by Smidth sparked interest in lensing, which then lead to the discovery of 0957+561 In 1979. (Review: https://arxiv.org/pdf/1304.3627.pdf

The introduction refers to only one publication. However, since it’s a review article, it would be beneficial to provide more citations to relevant work. 

Paragraphs starting at Line 662, it is worth mentioning here that Refsdal proposed this method of measuring the Hubble parameter to be applied to a supernova lying far behind and close to the line of sight through a distant galaxy, not quasars (https://academic.oup.com/mnras/article/128/4/307/2601707). It would also be interesting to mention that such supernovas haven’t been observed until 2014 (https://science.sciencemag.org/content/347/6226/1123). 

Section 13. Starts at Line 757. Please provide the name of the mentioned lensed systems.

Figure 11. Please provide the name of the mentioned lensed systems. 

Figure 13. What is the source of the image? 

In the last few sections, the review is focusing on the black hole shadow and other applications. As the authors have mentioned, it is impossible to include here all essential developments in the field. However, it would be very beneficial for readers to have a chance also to provide references to work related to gravitational microlensing of quasars which has been successfully used to study inner structures of AGNs (e.g. Irwin et al. 1989, Anguita et al. 2008, Morgan et al. 2008, Dai et al. 2010, Mosquera & Kochanek 2011). Such references would be very beneficial in the paragraph starting at line 1142, where microlensing of quasars is mentioned. However, no references are given. Even more, interestingly, the method will provide a significant boost of our understanding of the AGNs structure, as well as, black hole mass and spin thanks to the predicted increase of the number of such systems known. Today, we know a few dozens of microlensed quasars. By the mid-2020s, due to wide-field surveys including Dark Energy Survey, DESI, LSST, and Euclid, even up to 1000s of suitable systems will be found (see https://arxiv.org/pdf/1904.12967.pdf). 

An important development in the filed of lensing that has been omitted here that has is the use of the effect of light bending to resolve high energy universe, in particular, to resolve the structure of relativistic jets, infer the origin of gamma rays, and future applications for cosmology (see review https://www.sciencedirect.com/science/article/abs/pii/S0370157318302254). 

Gravitational lensing has been successfully used to study astrophysical phenomena for hundreds of sources. It would be very beneficial to end the article with emphasizes that the next decade will belong to gravitational lensing. 

 Future surveys, including LSST, SKA, and Euclid, will provide observations for hundreds of thousands of gravitationally lensed sources, which will allow us to apply gravitational lensing to study a variety of astrophysical problems. The future increase of three orders of magnitude in the number of lensed systems will turn the next decade into the era of strong gravitational lensing. The conclusion of the manuscript do not mention these upcoming revolution in astrophysics that will also bring the legacy of the Einstein eclipse to the next level.  

Author Response

Dear Universe Editors, We have made the amendments suggested by reviewer 2 and reviewer 4 (the other two reviewers had no criticisms). These are as follows:  

Reviewer 2:

Related to the first criticism: On lines 359-360 of the new version the phase “spacetime curvature in the presence of matter fields [26].” was changed to “both time and space contributions were now considered, whereas the first calculation took into account only the temporal curvature contribution [28]."

Related to the second criticism: On line 607 of the new version the phrase “...has no unique focal...” was changed to “...has not necessarily a unique focal…”, and in lines 613-617 we added “Though there are these differences, in practice one finds configurations that possess a well-defined focal plane, e.g. when one approximates a lens distribution as a sheet of uniform surface density, and in this case, the analogy to a lens is well approximated, except only for the spatial dependence of the lens refractive index. “

We corrected some typos and grammatical mistakes. We highlighted these changes in yellow, as well as some reference additions suggested by the editors.

Reviewer 4:

In general we agree with the comments, however, we opted for not providing a list references in the very short introduction, since otherwise we had to mention too many works that in fact are quoted in an historical order in the rest of the manuscript.

Related to the other criticisms, we changed (highlighted in green in the manuscript):

In line 46, the phase “strong magnification of faint sources” is replaced by “magnification of sources”.

In lines 56-59, we added that lensing can be used to resolve structures. The text now reads:

Lensing applications cover a wide variety of topics, among them it has been used in the search of exoplanets or to resolve structures of active galaxies. Gravitational lensing has been applied to the entire electromagnetic spectrum from radio to gamma rays, and recently using gravitational waves, allowing us to study structures as small as black holes and as large as galaxy clusters.

Related to the comment “Line 74: Measuring redshift of quasi-stellar object 3C 273 and concluding it’s cosmic origin in 1964 by Smidth sparked interest in lensing, which then lead to the discovery of 0957+561 In 1979. ”

We added the following text in lines 682-684:

…. It is worth mentioning that its identified cosmic origin sparked attention in lensing. Then, interest arose to study events that would lead to quasar identification [66].

Related to the comment “Paragraphs starting at Line 662, it is worth mentioning here that Refsdal proposed this method of measuring the Hubble parameter to be applied to a supernova lying far behind and close to the line of sight through a distant galaxy, not quasars (https://academic.oup.com/mnras/article/128/4/307/2601707). It would also be interesting to mention that such supernovas haven’t been observed until 2014 (https://science.sciencemag.org/content/347/6226/1123). ”

We added, in lines 678-680:

This method was thought to be applied to a supernova lying far behind and close to the line of sight through a distant galaxy, however, it was only until much later measured [64]. In that paper Refsdal also …

In line, 776, we provided the name of the quasar QSO 0957+561.

In Figure 11 we provided the name of the quasar and in figure 13 we wrote the name of  the image source.

Referring to the microlensing references, we included them in the new lines 1166 to 1172. It reads:

Also, this microlensing technique has been successfully used to study inner structures of AGNs, from the early evidence in 1989 [134] to the present [135,136,137,138,139]. It seems promising that the method will provide a significant boost of our understanding of the AGNs structure, as well as black hole mass and spin thanks to the predicted increase of the number of such systems known. Today, we know a few dozens of bright four-image microlensed quasars systems, but by the mid-2020s, due to wide-field surveys including DES, DESI, LSST, and Euclid, hundreds even up to thousands of suitable systems are expected to be found [140].

Finally, the last three comments, including  references, were considered in a paragraph in last, concluding section 19. Lines 1421-1432, it reads:

Future surveys will provide observations for a large number of gravitationally lensed sources, that will allow us to apply lensing to study a variety of astrophysical problems. The increase of a few orders of magnitude in the number of lensed systems will turn the next decade into an exciting era of gravitational lensing. A significant extension on the uses of strong lensing will be its practice as a method to infer precise locations of high energy emissions sources [190]. As we have already mentioned, gravitational lenses induce time delays and produce multiple images of sources. Both effects on lensed images can be used to drastically improve the angular resolution of current telescopes. In other words, gravitational lenses can in fact act as high-resolution telescopes. This will bring enormous consequences to astrophysics as will allow us to explore e.g. the inner regions of active galaxies. These regions host the most extreme and energetic phenomena in the universe including, relativistic jets, supermassive black hole binaries, and recoiling supermassive black holes. For an outlook on the future of this topic we refer the reader to a recent review [191].